# Distributional Successor Features Enable Zero-Shot Policy Optimization

**Chuning Zhu**
University of Washington
zchuning@cs.washington.edu

**Xinqi Wang**
University of Washington
wxqkaxdd@cs.washington.edu

**Tyler Han**
University of Washington
than123@cs.washington.edu

**Simon Shaolei Du**
University of Washington
ssdu@cs.washington.edu

**Abhishek Gupta**
University of Washington
abhgupta@cs.washington.edu

## Abstract

Intelligent agents must be generalists, capable of quickly adapting to various tasks. In reinforcement learning (RL), model-based RL learns a dynamics model of the world, in principle enabling transfer to arbitrary reward functions through planning. However, autoregressive model rollouts suffer from compounding error, making model-based RL ineffective for long-horizon problems. Successor features offer an alternative by modeling a policy's long-term state occupancy, reducing policy evaluation under new rewards to linear regression. Yet, policy optimization with successor features can be challenging. This work proposes *a novel class of models*, i.e., Distributional Successor Features for Zero-Shot Policy Optimization (DiSPOs), that learn a distribution of successor features of a stationary dataset's behavior policy, along with a policy that acts to realize different successor features within the dataset. By directly modeling long-term outcomes in the dataset, DiSPOs avoid compounding error while enabling a simple scheme for zero-shot policy optimization across reward functions. We present a practical instantiation of DiSPOs using diffusion models and show their efficacy as a new class of transferable models, both theoretically and empirically across various simulated robotics problems. Videos and code are available at https://weirdlabuw.github.io/dispo/.

## 1 Introduction

Reinforcement learning (RL) agents are ubiquitous in a wide array of applications, from language modeling [8] to robotics [22, 28]. Traditionally, RL has focused on the single-task setting, learning behaviors that maximize a specific reward function. However, for practical deployment, RL agents must be able to generalize across different reward functions within an environment. For example, a robot deployed in a household setting should not be confined to a single task such as object relocation but should handle various tasks, objects, initial and target locations, and path preferences.

This work addresses the challenge of developing RL agents that can broadly generalize to *any* task in an environment specified by a reward function. To achieve this type of generalization, we consider the paradigm of pretraining on an offline dataset of transitions and inferring optimal policies for downstream tasks from observing task-specific rewards. Since the target task is not revealed during

38th Conference on Neural Information Processing Systems (NeurIPS 2024).

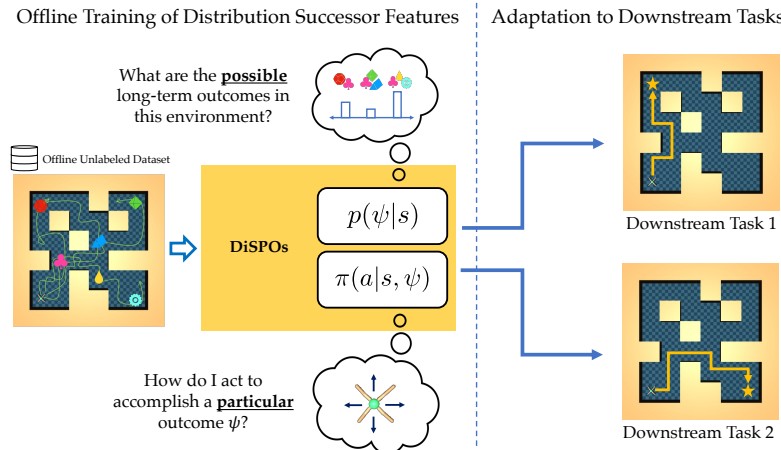

Figure 1: The transfer setting for DiSPOs. Given an unlabeled offline dataset, DiSPOs model both "what can happen?" $p(\psi|s)$ and "how can we achieve a particular outcome?" $p(a|s, \psi)$. This is used for quick adaptation to new downstream tasks without test-time policy optimization.

pretraining, the model must encode information about the environment dynamics without committing to a particular policy or reward. Moreover, once the task reward is observed, the model must provide a way to quickly evaluate and improve the policy since different tasks require different optimal policies.

A natural approach to this problem is model-based reinforcement learning [61, 21, 62], which learns an approximate dynamics model of the environment. Given a downstream reward function, task-optimal behavior can be obtained by "planning" via model rollouts [50, 58, 39, 45]. Typically, model rollouts are generated autoregressively, conditioning each step on generation from the previous step. In practice, however, autoregressive generation suffers from *compounding error* [31, 1, 26], which arises when small, one-step approximation errors accumulate over long horizons. This leads to rollout trajectories that diverge from real trajectories, limiting many model-based RL methods to short-horizon, low-dimensional problems.

An alternative class of algorithms based on *successor features* (SFs) has emerged as a potential approach to transferable decision-making [3, 2]. Successor features represent the discounted sum of features for a given policy. Assuming a linear correspondence between features and rewards, policy evaluation under new rewards reduces to a simple linear regression problem. Notably, by directly predicting long-term outcomes, SFs avoid autoregressive rollouts and hence compounding error. However, the notion of successor features is deeply tied to the choice of a particular policy. This policy dependence hinders the recovery of optimal policies for various downstream tasks. Current approaches to circumvent policy dependence either maintain a set of policies and select the best one during inference [3] or randomly sample reward vectors and make conditional policy improvements [5, 54, 55]. Nevertheless, a turnkey solution to transfer remains a desirable goal.

In this work, we propose a new class of models—***Distributional Successor Features for Zero-Shot Policy Optimization (DiSPOs)***—that are rapidly transferable across reward functions while avoiding compounding errors. Rather than modeling the successor features under a particular policy, DiSPOs model the *distribution* of successor features under the *behavior policy*, effectively encoding all possible outcomes that appear in the dataset at each state. Crucially, by representing outcomes as successor features, we enjoy the benefit of zero-shot outcome evaluation after solving a linear reward regression problem, without the pitfall of compounding error. In addition to the outcome model, DiSPOs jointly learn a readout policy that generates an action to accomplish a particular outcome. Together, these models enable zero-shot policy optimization [55] for arbitrary rewards without further training: at test time, we simply perform a linear regression, select the best in-distribution outcome, and query the readout policy for an action to realize it. *DiSPOs are a new class of world models as they essentially capture the dynamics of the world and can be used to plan for optimal actions under arbitrary rewards, without facing the pitfalls of compounding error.*

Since multiple outcomes can follow from a particular state, and multiple actions can be taken to achieve a particular outcome, both the outcome distribution and the policy require expressive model

classes to represent. We provide a practical instantiation of DiSPOs using diffusion models [23, 47] and show that under this parameterization, policy optimization can be cast as a variant of guided diffusion sampling [13]. We validate the transferability of DiSPOs across a suite of long-horizon simulated robotics domains and further show that DiSPOs provably converge to "best-in-data" policies. With DiSPOs, we hope to introduce a new way for the research community to envision transfer in reinforcement learning, and to think of alternative ways to address the challenges of world modeling.

## 2  Related Work

Our work has connections to numerous prior work on model-based RL and successor features.

**Model-Based RL** To enable transfer across rewards, model-based RL learns one-step (or multi-step) dynamics models via supervised learning and use them for planning [11, 38, 39, 58] or policy optimization [12, 50, 26, 64, 20]. These methods typically suffer from compounding error, where autoregressive model rollouts lead to large prediction errors over time [1, 31]. Despite improvements to model architectures [20, 25, 1, 32, 65] and learning objectives [27], modeling over long horizons without compounding error remains an open problem. DiSPOs instead directly model cumulative long-term outcomes in an environment, avoiding autoregressive generation while remaining transferable.

**Successor Features** Successor features achieve generalization across rewards by modeling the accumulation of features (as opposed to rewards in model-free RL) [3, 2]. With the assumption that rewards are linear in features, policy evaluation under new rewards reduces to a linear regression problem. A key limitation of successor features is their inherent policy dependence, as they are defined as the accumulated features when acting according to a particular policy. This makes extracting optimal policies for new tasks challenging.

To circumvent this policy dependence, generalized policy improvement [3, 2] maintains a discrete set of policies and selects the highest valued one to execute at test time, limiting the space of available policies for new tasks. Universal SF [6] and Forward-Backward Representations [54, 55] randomly sample reward weights $z$ and jointly learn successor features and policies conditioned on $z$. The challenge lies in achieving coverage over the space of all possible policies through sampling of $z$, resulting in potential distribution shifts for new problems. RaMP [9] learns a successor feature predictor conditioned on an initial state and a sequence of actions. Transfer requires planning by sampling actions sequences, which becomes quickly intractable over horizon. In contrast, DiSPOs avoid conditioning on any explicit policy representation by modeling the distribution of all possible outcomes represented in a dataset, and then selecting actions corresponding to the most desirable long-term outcome.

Distributional Successor Measure (DSM) [59] is a concurrent work that learns a distribution over successor representations using tools from distributional RL [4]. Importantly, DSM models the distributional successor measure of a *particular* policy, where the stochasticity stems from the policy and the dynamics. This makes it suitable for robust policy evaluation but not for transferring to arbitrary downstream tasks. In contrast, DiSPOs model the distribution of successor feature outcomes in the dataset (i.e., the behavior policy), where the distribution stems from the range of meaningfully distinct long-term outcomes. This type of modeling allows DiSPOs to extract optimal behavior for arbitrary downstream tasks, while DSMs suffer from the same policy dependence that standard successor feature-based methods do.

## 3  Preliminaries

We adopt the standard Markov Decision Process (MDP) notation and formalism [24] for an MDP $\mathcal{M} = (\mathcal{S}, \mathcal{A}, r, \gamma, \mathcal{T}, \rho_0)$, but restrict our consideration to the class of deterministic MDPs. While this does not encompass every environment, it does capture a significant set of problems of practical interest. Hereafter, we refer to a deterministic MDP and a *task* interchangeably. In our setting, we consider transfer across different tasks that always share the same action space $\mathcal{A}$, state space $\mathcal{S}$, and transition dynamics $\mathcal{T} : \mathcal{S} \times \mathcal{A} \rightarrow \mathcal{S}$[1] The difference between tasks only lies in having different state-dependent Markovian reward functions $r : \mathcal{S} \rightarrow [0, 1]$.

---

[1]For simplicity, we also use $\mathcal{T}(s, a)$ to denote the next state.

**Value Functions and Successor Features** Let $R = \sum_{t=1}^{\infty} \gamma^{t-1} r(s_t)$ denote the cumulative reward for a trajectory $\{s_i, a_i\}_{i=1}^{\infty}$. One can then define the state value function under policy $\pi$ as $V^\pi(s) := \mathbb{E}_{\pi,\mathcal{T}}[R \mid s_1 = s]$, and the state-action value function as $Q^\pi(s, a) := \mathbb{E}_{\pi,\mathcal{T}}[R \mid s_1 = s, a_1 = a]$. The value function admits a temporal structure that allows it to be estimated using dynamic programming, which iteratively applies the Bellman operator until a fixed point is reached $V^\pi(s) := r(s) + \gamma \mathbb{E}_\pi[V^\pi(s_2) \mid s_1 = s]$. While these Bellman updates are in the tabular setting, equivalent function approximator variants (e.g., with neural networks) can be instantiated to minimize a Bellman "error" with stochastic optimization techniques [37, 19, 36].

Successor features [2] generalize the notion of a value function from task-specific rewards to task-agnostic features. Given a state feature function $\phi : S \to \mathbb{R}^d$, the successor feature of a policy is defined as $\psi^\pi(s) = \mathbb{E}_{\pi,\mathcal{T}}\left[\sum_{t=1}^{\infty} \gamma^{t-1} \phi(s_i) \mid s_1 = s\right]$. Suppose rewards can be linearly expressed by the features, i.e. there exists $w \in \mathbb{R}^n$ such that $R(s) = w^\top \phi(s)$, then the value function for the particular reward can be linearly expressed by the successor feature $V^\pi(s) = w^\top \psi^\pi(s)$. Hence, given the successor feature $\psi^\pi$ of a policy $\pi$, we can immediately compute its value under any reward once the reward weights $w$ are known. Analogous to value functions, successor features also admit a recursive Bellman identity $\psi^\pi(s) := \phi(s) + \gamma \mathbb{E}_\pi[\psi^\pi(s')]$, allowing them to be estimated using dynamic programming [3]. In this paper, we also refer to the discounted sum of features along a *trajectory* as a successor feature. In this sense, a successor feature represents an outcome that is feasible under the dynamics and can be achieved by some policy.

**Diffusion Models** DiSPOs rely on expressive generative models to represent the distribution of successor features. Diffusion models [23, 47] are a class of generative models where data generation is formulated as an iterative denoising process. Specifically, DDPM [23] consists of a forward process that iteratively adds Gaussian noise to the data, and a corresponding reverse process that iteratively denoises a unit Gaussian to generate samples from the data distribution. The reverse process leverages a neural network estimating the score function of each noised distribution, trained with a denoising score matching objective [49]. In addition, one can sample from the conditional distribution $p(x|y)$ by adding a guidance $\nabla_x \log p(y|x)$ to the score function in each sampling step [13]. As we show in Sec. 4.3, guided diffusion enables quick selection of optimal outcomes from DiSPOs.

**Problem setting** We consider a transfer learning scenario with access to an offline dataset $\mathcal{D} = \{(s_i, a_i, s_i')\}_{i=0}^{N}$ of transition tuples collected with some behavior policy $\pi_\beta$ under dynamics $\mathcal{T}$. The goal is to quickly obtain the optimal policy $\pi^*$ for some downstream task, specified in the form of a reward function or a number of $(s, r)$ samples. While we cannot hope to extrapolate beyond the dataset (as is common across problems in offline RL [34]), we will aim to find the best policy *within* dataset coverage for the downstream task. This is defined more precisely in Section 4.2.

## 4 Distributional Successor Features for Zero-Shot Policy Optimization

We introduce the framework of DiSPOs as a scalable approach to the transfer problem described in Section 3, with the goal of learning from an unlabeled dataset to quickly adapt to any downstream task specified by a reward function. We start by relating the technical details behind learning DiSPOs in Section 4.1, followed by explaining how DiSPOs can be used for efficient multi-task transfer in Section 4.2. Finally, we describe a practical instantiation of DiSPOs in Section 4.3.

### 4.1 Learning Distributional Successor Features of the Behavior Policy

To transfer and obtain optimal policies across different reward functions, generalist decision-making agents must model the future in a way that permits the evaluation of new rewards *and* new policies. To this end, DiSPOs adopt a technique based on off-policy dynamic programming to directly model the distribution of cumulative future outcomes, without committing to a particular reward function $r(\cdot)$ or policy $\pi$. Fig. 2 illustrates the two components in DiSPOs, and we describe each below.

**(1) Outcome model:** for a particular a state feature function $\phi(s)$, DiSPOs model the distribution of successor features $p(\psi|s)$ over all paths that have coverage in the dataset. In deterministic MDPs, each successor feature $\psi$ (discounted sum of features $\psi = \sum_t \gamma^{t-1} \phi(s_t)$) can be regarded as an "outcome". When the state features are chosen such that reward for the desired downstream task is a linear function of features, i.e., there exists $w \in \mathbb{R}^n$ such that $r(s) = w^\top \psi(s)$ [42, 43, 66, 56, 9, 3], the value of each outcome can be evaluated as $w^\top \psi$. That is, knowing $w$ effectively transforms the

distribution of outcomes $p(\psi|s)$ into a distribution of task-specific values (sum of rewards) $p(R|s)$. Notably, since $w$ can be estimated by regressing rewards from features, distributional evaluation on a new reward function boils down to a simple linear regression.

As in off-policy RL, the outcome distribution $p(\psi|s)$ in DiSPOs can be learned via an approximate dynamic programming update, which is similar to a distributional Bellman update [4]:

$$\max_{\theta} \ \mathbb{E}_{(s,a,s')\sim\mathcal{D}} \left[\log p_{\theta}(\phi(s) + \gamma\psi_{s'}|s)\right]$$
$$\text{s.t} \qquad \psi_{s'} \sim p_{\theta}(\cdot|s') \tag{1}$$

Intuitively, this update suggests that the distribution of successor features $p_{\theta}(\psi|s)$ at state $s$ maximizes likelihood over current state feature $\phi(s)$ added to sampled future outcomes $\psi_{s'}$. This instantiates a fixed-point procedure, much like a distributional Bellman update. An additional benefit of the dynamic programming procedure is trajectory stitching, where combinations of subtrajectories in the dataset will be represented in the outcome distribution.

**(2) Readout policy:** Modeling the distribution of future outcomes in an environment is useful only when it can be realized in terms of actions that accomplish particular outcomes. To do so, DiSPOs pair the outcome model with a readout policy $\pi(a|s,\psi)$ that actualizes a desired long-term outcome $\psi$ into the action $a$ to be taken at state $s$. Along with the outcome model $p_{\theta}(\psi|s)$, the readout policy $\pi_{\rho}(a|s,\psi)$ can be optimized via maximum-likelihood estimation:

$$\max_{\rho} \ \mathbb{E}_{(s,a,s')\sim\mathcal{D}} \left[\log \pi_{\rho}(a|s,\psi = \phi(s) + \gamma\psi_{s'})\right]$$
$$\text{s.t} \qquad \psi_{s'} \sim p_{\theta}(.|s') \tag{2}$$

This update states that if an action $a$ at a state $s$ leads to a next state $s'$, then $a$ should be taken with high likelihood for outcomes $\psi$, which are a combination of the current state feature $\phi(s)$ and future outcomes $\psi_{s'} \sim p_{\theta}(\cdot|s')$.

The outcome distribution $p(\psi|s)$ can be understood as a natural analogue to a value function, but with two crucial differences: (1) it represents the accumulation of not just a single reward function but an arbitrary feature (with rewards being linear in this feature space), and (2) it is not specific to any particular policy but represents the distribution over all cumulative outcomes covered in the dataset. The first point enables transfer across rewards, while the second enables the selection of optimal actions for new rewards rather than being restricted to a particular (potentially suboptimal) policy. Together with the readout policy $\pi(a|s,\psi)$, these models satisfy our desiderata for transfer, i.e., that the value for new tasks can be estimated by simple linear regression without requiring autoregressive generation, and that optimal actions can be obtained without additional policy optimization.

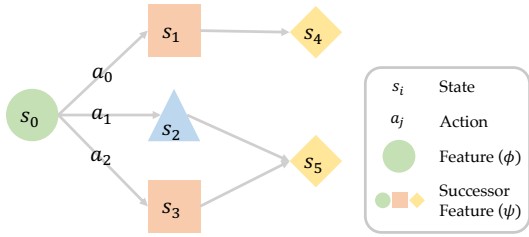

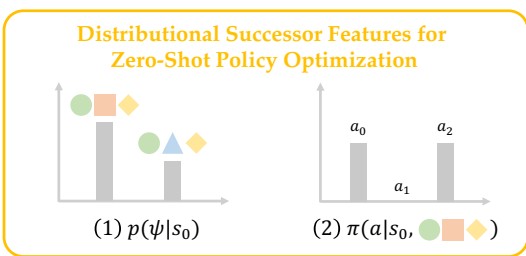

Figure 2: DiSPOs for a simple environment. Given a state feature function $\phi$, DiSPOs learn a distribution of all possible long-term outcomes (successor features $\psi$) in the dataset $p(\psi|s)$, along with a readout policy $\pi(a|s,\psi)$ that takes an action $a$ to realise $\psi$ starting at state $s$.

## 4.2 Zero-Shot Policy Optimization with Distributional Successor Features

To synthesize optimal policies for novel downstream reward functions using DiSPOs, two sub-problems must be solved: (1) inferring the suitable linear reward weights $w_r$ for a particular reward function from a set of $(s, r)$ tuples and (2) using the inferred $w_r$ to select an optimal action $a^*$ at a state $s$. We discuss each below.

**Inferring task-specific weights with linear regression.** As noted, for any reward function $r(s)$, once the linear reward weights $w_r$ are known (i.e., $r(s) = w_r^T \phi(s)$), the distribution of returns in the dataset $p(R|s)$ is known through linearity. However, in most cases, rewards are not provided in

functional form, making $w_r$ unknown a priori. Instead, given a dataset of $\mathcal{D} = \{(s, r)\}$ tuples, $w_r$ can be obtained by solving a simple linear regression problem $\arg\min_{w_r} \frac{1}{|\mathcal{D}|} \sum_{(s,r) \in \mathcal{D}} \|w_r^T \psi(s) - r\|_2^2$.

**Generating task-specific policies via distributional evaluation.** Given the inferred $w_r$ and the corresponding future return distribution $p(R|s)$ obtained through linear scaling of $p(\psi|s)$, the optimal action can be obtained by finding the $\psi$ with the highest possible future return that has sufficient data-support:

$$\psi^* \leftarrow \arg\max_{\psi} \quad w_r^T \psi, \qquad \text{s.t} \quad p(\psi|s) \geq \epsilon, \tag{3}$$

where $\epsilon > 0$ is a hyperparameter to ensure sufficient coverage for $\psi$. This suggests that the optimal outcome $\psi^*$ is the one that provides the highest future sum of rewards $w_r^T \psi^*$ while being valid under the environment dynamics and dataset coverage.

This optimization problem can be solved in a number of ways. The most straightforward is via random shooting [53], which samples a set of $\psi$ from $p(\psi|s)$ and chooses the one with the highest $w_r^T \psi$. Sec. 5 bases our theoretical analysis on this technique. Sec. 4.3 shows that for outcome models instantiated with diffusion models, the optimization problem can be simplified to guided diffusion sampling.

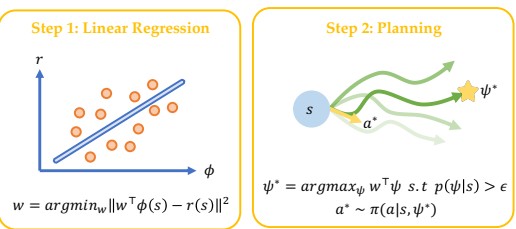

Figure 3: Zero-shot policy optimization with DiSPOs. Once a DiSPO is learned, the optimal action can be obtained by performing reward regression and searching for the optimal outcome under the dynamics to decode via the policy.

Once $\psi^*$ has been obtained, the action to execute in the environment can be acquired via the readout policy $\pi_\rho(a|s, \psi^*)$. Fig. 3 shows the full policy optimization procedure, and we refer the reader to Appendix. F for the pseudocode.

As described previously, DiSPOs enable zero-shot transfer to arbitrary new rewards in an environment without accumulating compounding error or requiring expensive test-time policy optimization. In this way, they can be considered a new class of models of transition dynamics that avoids the typical challenges in model-based RL and successor features.

## 4.3 Practical Instantiation

In this section, we provide a practical instantiation of DiSPOs that is used throughout our experimental evaluation. The first step to instantiate DiSPOs is to choose an expressive state feature that linearly expresses a broad class of rewards. We choose the state feature $\phi$ to be $d$-dimensional random Fourier features [52]. Next, the model class must account for the multimodal nature of outcomes and actions since multiple outcomes can follow from a state, and multiple actions can be taken to realize an outcome. To this end, we parametrize both the outcome model $p(\psi|s)$ and the readout policy $\pi(a|s, \psi)$ using a conditional diffusion model [23, 47]. We then train these models (optimize Equation 1, 2) by denoising score matching, a surrogate of maximum likelihood training [48].

Remarkably, when $p(\psi|s)$ is parameterized by a diffusion model, the special structure of the optimization problem in Eq. 3 allows a simple variant of guided diffusion [14, 25] to be used for policy optimization. In particular, taking the log of both sides of the constraint and recasting the constrained optimization via the penalty method, we get a penalized objective $\mathcal{L} = w_r^T \psi + \alpha(\log p(\psi|s) - \log \epsilon)$. Taking the gradient yields

$$\nabla_\psi \mathcal{L}(\psi, \alpha) = w_r + \alpha \nabla_\psi \log p(\psi|s). \tag{4}$$

The expression for $\nabla_\psi \mathcal{L}(\psi, \alpha)$ is simply the score function $\nabla_\psi \log p(\psi|s)$ in standard diffusion training (Section 3), with the linear weights $w_r$ added as a guidance term. Planning then becomes doing stochastic gradient Langevin dynamics [57] to obtain an optimal $\psi^*$ sample, using $\nabla_\psi \mathcal{L}(\psi, \alpha)$ as the gradient. Guided diffusion removes the need for sampling a set of particles. As shown in Appendix E, it matches the performance of random shooting while taking significantly less inference time. In Appendix B, we show that the guided diffusion procedure can alternatively be viewed as taking actions conditioned on a soft optimality variable.

# 5 Theoretical Analysis of Distributional Successor Features for Zero-Shot Policy Optimization

To provide a theoretical understanding of DiSPOs, we conduct an error analysis to connect the error in estimating the ground truth $p_0(\psi \mid s)$ to the suboptimality of the DiSPO policy, and then study when the DiSPO policy becomes optimal. We start our analysis conditioning on $\epsilon > 0$ estimation error in the ground truth outcome distribution $p_0(\psi \mid s)$.

**Condition 5.1.** We say the learnt outcome distribution $\hat{p}$ is an $\epsilon$-good approximation if $\forall\, s \in \mathcal{S}$, $\|\hat{p}(\psi \mid s) - p_0(\psi \mid s)\|_\infty \leq \epsilon$.

Since DiSPOs capture the outcome distribution of the behavior policy $\pi_\beta$, we need a definition to evaluate a policy $\pi$ with respect to $\pi_\beta$.

**Definition 5.2.** We say a state-action pair $(s,a)$ is $(\delta, \pi_\beta)$-good if over the randomness of $\pi_\beta$, $\mathbb{P}_{\pi_\beta}[Q^{\pi_\beta}(s,a) < \sum_{t=1}^\infty \gamma^{t-1} r(s_t) \mid s_1 = s] \leq \delta$. Furthermore, if for all state $s$, $(s, \pi(s))$ is $(\delta, \pi_\beta)$-good, then we call $\pi$ a $(\delta, \pi_\beta)$-good policy.

We proceed to use Definition 5.2 to characterize the suboptimality of the DiSPO policy. Let $\tau$ denote the sampling optimality of the random shooting planner in Sec. 4.2. Specifically, we expect to sample a top $\tau$ outcome $\psi$ from the behavior policy in $O(\frac{1}{\tau})$ samples, where $\mathbb{P}_{\pi_\beta}[w_r^T \psi \leq \sum_t \gamma^{t-1} r(s_t)] \leq \tau$. The following result characterizes the suboptimality of the DiSPO policy. The proof is deferred to Appendix. A.

**Theorem 5.3** (main theorem). *For any MDP $\mathcal{M}$ and $\epsilon$-good outcome distribution $\hat{p}$, the policy $\hat{\pi}$ given by the random shooting planner with sampling optimality $\tau$ is a $(\epsilon + \tau, \pi_\beta)$-good policy.*

From Theorem 5.3, we can obtain the following suboptimality guarantee in terms of the value function under the Lipschitzness condition. The corollary shows the estimation error in $p_0(\psi \mid s)$ will be amplified by an $O\left(\frac{1}{1-\gamma}\right)$ multiplicative factor.

**Corollary 5.4.** *If we have $\lambda$-Lipschitzness near the optimal policy, i.e., $Q^*(s, a^*) - Q^*(s,a) \leq \lambda\delta$ when $(s,a)$ is $(\delta, \beta)$-good, the suboptimality of output policy $\hat{\pi}$ is $V_0^*(s_0) - V_0^{\hat{\pi}}(s_0) \leq \frac{\lambda}{1-\gamma}(\tau + \epsilon)$.*

Lastly, we extend our main theoretical result to the standard full data coverage condition in the offline RL literature, where the dataset contains all transitions [51, 60, 44]. The following theorem states that DiSPOs can output the optimal policy in this case. The proof is deferred to Appendix A.

**Theorem 5.5.** *In deterministic MDPs, when $|\mathcal{A} \times \mathcal{S}| < \infty$, and $\forall (s,a) \in \mathcal{S} \times \mathcal{A}, N(s, a, \mathcal{T}(s,a)) \geq 1$, DiSPOs are guaranteed to identify an optimal policy.*

# 6 Experimental Evaluation

In our experimental evaluation, we aim to answer the following research questions. (1) Can DiSPOs transfer across tasks without expensive test-time policy optimization? (2) Can DiSPOs avoid the challenge of compounding error present in model-based RL? (3) Can DiSPOs solve tasks with arbitrary rewards beyond goal-reaching problems? (4) Can DiSPOs go beyond the offline dataset, and accomplish "trajectory-stitching" to actualize outcomes that combine different subtrajectories?

We answer these questions through a number of experimental results in simulated robotics problems. We defer detailed descriptions of domains and baselines to Appendix D and C, as well as detailed ablative analysis to Appendix E.

## 6.1 Problem Domains and Datasets

**Antmaze** [15] is a navigation domain that involves controlling a quadruped to reach some designated goal location. Each task corresponds to reaching a different goal location. We use the D4RL dataset for pretraining and dense rewards described in Appendix D for adaptation.

**Franka Kitchen** [15] is a manipulation domain where the goal is to control a Franka arm to interact with appliances in the kitchen. Each task corresponds to interacting with a set of items. We use the D4RL dataset for pretraining and standard sparse rewards for adaptation.

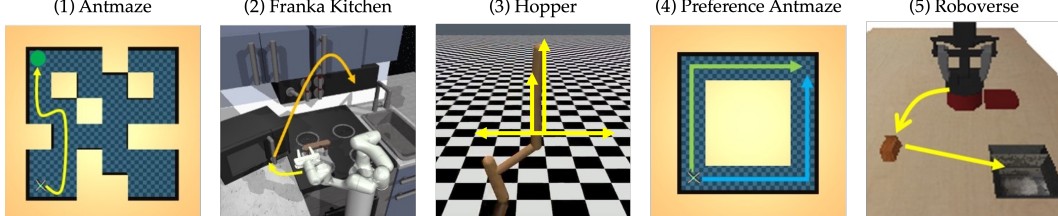

Figure 4: Evaluation domains: (1) D4RL Antmaze [15] (2) Franka Kitchen [15] (3) Hopper [9] (4) Preference-Based Antmaze with the goal of taking a particular path (5) Roboverse [46] robotic manipulation.

Table 1: Offline multitask RL on AntMaze and Kitchen. DiSPOs show superior transfer performance (in average episodic return) than successor features, model-based RL, and misspecified goal-conditioned baselines.

|  | DiSPO (Ours) | USFA | FB | RaMP | MOPO | COMBO | GC-IQL |
|---|---|---|---|---|---|---|---|
| umaze | **593** $\pm$ **16** | 462 $\pm$ 4 | 469 $\pm$ 12 | 459 $\pm$ 3 | 451 $\pm$ 2 | 574 $\pm$ 10 | 571 $\pm$ 15 |
| umaze-diverse | **568** $\pm$ **12** | 447 $\pm$ 3 | 474 $\pm$ 2 | 460 $\pm$ 7 | 467 $\pm$ 5 | 547 $\pm$ 11 | 577 $\pm$ 7 |
| medium-diverse | **631** $\pm$ **67** | 394 $\pm$ 52 | 294 $\pm$ 61 | 266 $\pm$ 2 | 236 $\pm$ 4 | 418 $\pm$ 16 | 403 $\pm$ 10 |
| medium-play | **624** $\pm$ **58** | 370 $\pm$ 31 | 264 $\pm$ 29 | 271 $\pm$ 5 | 232 $\pm$ 4 | 397 $\pm$ 12 | 390 $\pm$ 33 |
| large-diverse | **359** $\pm$ **59** | 215 $\pm$ 20 | 181 $\pm$ 46 | 132 $\pm$ 1 | 128 $\pm$ 1 | 244 $\pm$ 19 | 226 $\pm$ 9 |
| large-play | **306** $\pm$ **18** | 250 $\pm$ 41 | 165 $\pm$ 12 | 134 $\pm$ 3 | 128 $\pm$ 2 | 248 $\pm$ 4 | 229 $\pm$ 5 |
| kitchen-partial | **43** $\pm$ **6** | 0 $\pm$ 0 | 4 $\pm$ 4 | 0 $\pm$ 0 | 8 $\pm$ 7 | 11 $\pm$ 9 | - |
| kitchen-mixed | **46** $\pm$ **5** | 10 $\pm$ 10 | 5 $\pm$ 5 | 0 $\pm$ 0 | 0 $\pm$ 0 | 0 $\pm$ 0 | - |
| hopper-forward | 566 $\pm$ 63 | 487 $\pm$110 | 452 $\pm$ 59 | 470 $\pm$ 16 | 493 $\pm$114 | **982** $\pm$**157** | - |
| hopper-backward | 367 $\pm$ 15 | 261 $\pm$ 68 | 269 $\pm$ 77 | 220 $\pm$ 15 | **596** $\pm$**211** | 194 $\pm$ 74 | - |
| hopper-stand | **800** $\pm$ **0** | 685 $\pm$130 | 670 $\pm$120 | 255 $\pm$ 15 | **800** $\pm$ **0** | 600 $\pm$111 | - |
| hopper-jump | **832** $\pm$ **22** | 746 $\pm$112 | 726 $\pm$ 35 | 652 $\pm$ 28 | 753 $\pm$ 51 | 670 $\pm$109 | - |

**Hopper** [7, 9] is a locomotion domain that involves controlling a hopper to perform various tasks, including hopping forward, hopping backward, standing, and jumping. We use the offline dataset from [9] for pretraining and shaped rewards for adaptation.

**Preference Antmaze** is a variant of D4RL Antmaze [15] where the goal is to reach the top right corner starting from the bottom left corner. The two tasks in this environment correspond to the two paths to reach the goal, simulating human preferences. We collect a custom dataset and design reward functions for each preference.

**Roboverse** [46] is a tabletop manipulation environment with a robotic arm completing multi-step problems. Each task consists of two phases, and the offline dataset contains separate trajectories of each phase but not full task completion. A sparse reward is assigned to each time step of task completion.

## 6.2 Baseline Comparisons

**Successor Features** We compare with three methods from the successor feature line of work. **USFA** [6] overcomes the policy dependence of SF by randomly sampling reward weights $z$ and jointly learning a successor feature predictor $\psi_z$ and a policy $\pi_z$ conditioned on $z$. $\psi_z$ captures the successor feature of $\pi_z$, while $\pi_z$ is trained to maximize the reward described by $z$. **FB** [54, 55] follows the same paradigm but jointly learns a feature network by parameterizing the successor measure as an inner product between a forward and a backward representation. **RaMP** [9] removes the policy dependence of SF by predicting cumulative features from an initial state and an open-loop sequence of actions, which can be used for planning.

**Model-Based RL** We compare with two variants of model-based reinforcement learning. **MOPO** [64] is a model-based offline RL method that learns an ensemble of dynamics models and performs actor-critic learning. **COMBO** [63] introduces pessimism into MOPO by training the policy using a conservative objective [30].

Table 2: Evaluation on non-goal-conditioned tasks. DiSPOs are able to solve non-goal-conditioned tasks, taking different paths in preference antmaze (Fig 4), while goal-conditioned RL cannot optimize for arbitrary rewards.

|       | DiSPO (Ours) | COMBO      | GC-IQL     |
|-------|--------------|------------|------------|
| Up    | $139 \pm 1$  | $143 \pm 9$ | $72 \pm 19$ |
| Right | $\mathbf{142} \pm 2$ | $136 \pm 4$ | $83 \pm 25$ |

Table 3: Evaluation of trajectory stitching ability of DiSPOs. DiSPOs outperform non-stitching baselines, demonstrating their abilities to recombine outcomes across trajectory segments

|              | DiSPO (Ours) | RaMP      | DT        |
|--------------|--------------|-----------|-----------|
| PickPlace    | $\mathbf{49} \pm 8$ | $0 \pm 0$ | $0 \pm 0$ |
| ClosedDrawer | $\mathbf{40} \pm 5$ | $0 \pm 0$ | $0 \pm 0$ |
| BlockedDrawer| $\mathbf{66} \pm 7$ | $0 \pm 0$ | $0 \pm 0$ |

**Goal-Conditioned RL** Goal-conditioned RL enables adaptation to multiple downstream goals $g$. However, it is solving a more restricted class of problems than RL as goals are less expressive than rewards in the same state space. Moreover, standard GCRL is typically trained on the same set of goals as in evaluation, granting them privileged information. To account for this, we consider a goal-conditioned RL baseline **GC-IQL** [40, 29] and only train on goals from half the state space to show its fragility to goal distributions. We include the original method trained on test-time goals in Appendix E.

### 6.3 Do DiSPOs enable zero-shot policy optimization across tasks?

We evaluate DiSPOs on transfer problems, where the dynamics are shared, but the reward functions vary. We train DiSPOs on the data distributions provided with the D4RL [15] and Hopper datasets. We identify the test-time reward by subsampling a small number of transitions from the offline dataset, relabeling them with the test-time rewards, and performing linear least squares regression. While DiSPOs in principle can identify the task reward from online experience, we evaluate in the offline setting to remove the confounding factor of exploration.

Table 1 reports the episodic return on D4RL and Hopper tasks. DiSPOs are able to transfer to new tasks with no additional training, showing

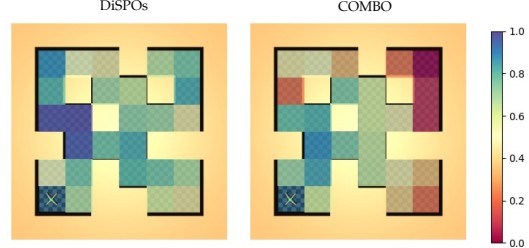

Figure 5: Transfer across tasks with DiSPOs and COMBO [63] in medium antmaze. Each tile corresponds to a different task, with color of the tile indicating the normalized return. DiSPOs successfully transfer across a majority of tasks, while MBRL [63] struggles on tasks that are further away from the initial location.

significantly higher performance than successor features (mismatch between training and evaluation policy sets), model-based RL (compounding error) and goal-conditioned RL (goal distribution misspecification). Notably, we show in Appendix E that DiSPOs are even competitive with goal-conditioned RL methods trained on test-time goals. The transferability of DiSPOs can also be seen in Fig 5, where we plot the performance of DiSPOs across various tasks (corresponding to different tiles in the maze). We see that DiSPOs have less degradation across tasks than model-based RL [63].

Although the DiSPO framework and theoretical results are derived under deterministic MDPs, we emphasize that the D4RL antmaze datasets are collected with action noise, emulating stochastic transitions. These results indicate that DiSPOs are practically applicable to some range of stochastic settings, although we expect it to perform better in purely deterministic settings.

### 6.4 Can DiSPOs solve tasks with *arbitrary* rewards?

While methods like goal-conditioned RL [40, 17] are restricted to shortest path goal-reaching problems, DiSPOs are able to solve problems with *arbitrary* reward functions. This is crucial when the reward is not easily reduced to a particular "goal". To validate this, we evaluate DiSPOs on tasks that encode nontrivial human preferences in a reward function, such as particular path preferences in antmaze. In this case, we have different rewards that guide the agent specifically down the path to the left and the right, as shown in Fig 4. As we see in Table 2, DiSPOs and model-based RL obtain policies that respect human preferences and are performant for various rewards. Goal-conditioned algorithms are unable to disambiguate preferences and end up with some probability of taking each path.

### 6.5 Do DiSPOs perform trajectory stitching?

The ability to recover optimal behavior by combining suboptimal trajectories, or "trajectory stitching," is crucial to off-policy RL methods as it ensures data efficiency and avoids requirements for exponential data coverage. DiSPOs naturally enables this type of trajectory stitching via the distributional Bellman backup, recovering "best-in-data" policies for downstream tasks. To evaluate the ability of DiSPOs to perform trajectory stitching, we consider the environments introduced in [46]. Here, the data only consists of trajectories that complete individual subtasks (e.g. grasping or placing), while the task of interest rewards the completion of both subtasks. Since the goal of this experiment is to evaluate stitching, not transfer, we choose the features as the task rewards $\phi(s) = r(s)$. We find that DiSPOs are able to show non-trivial success rates by stitching together subtrajectories. Since RaMP [9] predicts the summed features from a sequence of actions, and the optimal action sequence is not present in the dataset, it fails to solve any task. Likewise, return-conditioned supervised learning methods like Decision Transformer [10] do not stitch together trajectories and fails to learn meaningful behaviors.

## 7 Discussion

This work introduced Distributional Successor Features for Zero-Shot Policy Optimization (DiSPOs), a method for transferable reinforcement learning that does not incur compounding error or test-time policy optimization. By modeling the distribution of *all* possible future outcomes along with policies to reach them, DiSPOs can quickly provide optimal policies for *any* reward in a zero-shot manner. We presented an efficient algorithm to learn DiSPOs and demonstrated the benefits of DiSPOs over standard successor features and model-based RL techniques. The limitations of our work open future research opportunities. First, DiSPOs require a choice of features $\phi(s)$ that linearly express the rewards; this assumption may fail, necessitating more expressive feature learning methods. Second, DiSPOs model the behavior distribution of the dataset; hence, policy optimality can be affected by dataset skewness, which motivates the use of more efficient exploration methods for data collection. Finally, the current version of DiSPOs infer the reward from offline state-reward pairs; a potential future direction could apply this paradigm to online adaptation, where the reward is inferred from online interactions.

## Acknowledgment

CZ is supported by the UW-Amazon fellowship. TH is supported by the NSF GRFP under Grant No. DGE 2140004. SSD acknowledges the support of NSF IIS 2110170, NSF DMS 2134106, NSF CCF 2212261, NSF IIS 2143493, NSF CCF 2019844, and NSF IIS 2229881.

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

# Supplementary Materials for
# "Distributional Successor Features Enable Zero-Shot Policy Optimization"

## A    Missing Proofs

We provide the complete proofs of theorems and corollaries stated in Sec. 5. Throughout the following sections, we use $1_{\mathcal{E}}$ to denote the indicator of event $\mathcal{E}$.

### A.1    Formal Statement and Proof of Theorem 5.3

To state Theorem 5.3 rigorously, we introduce the basic setting here. Without loss of generality, let the feature at time step $i$, $\phi(s_i) \in [0, 1-\gamma]^d$ and the outcome $\psi = \sum_{i=0}^{\infty} \gamma^i \phi(s_i) \in [0,1]^d$. Moreover, we have a readout policy $\pi$ s.t. $\hat{a} \sim \hat{\pi}(s, \psi)$ always leads to a successor state $s'$ s.t. $\hat{p}(\frac{1}{\gamma}(\psi - \phi(s)) \mid s') > 0$.

We simplify the policy optimization phase of DiSPOs into the following form: for a given reward weight $w_r$, in each time step, we have

1. Infer optimal outcome $\psi^* = A(s, \hat{p})$ through random shooting;

2. Get corresponding action from $\hat{\pi}$, $\hat{a} \sim \hat{\pi}(s, \psi^*)$.

where the random shooting oracle A with sampling optimality $\tau$ satisfies

$$w_r^T A(s, \hat{p}) \geq \min\{R \mid \int 1_{[w_r^T \psi \geq R]} \hat{p}[\psi \mid s] d\psi \leq \tau\}.$$

This intuitively means that there are at most probability $\tau \in [0, 1]$ of the behavior policy achieving higher reward. We proceed to prove Theorem 5.3.

*Proof.* It suffices to prove that $\forall s, (s, \hat{\pi}(s))$ is at least $(\tau + \epsilon, \pi_\beta)$-good. For simplicity, we denote $\widehat{R} := \min\{R \mid \int 1_{[w_r^T \psi \geq R]} \hat{p}[\psi \mid s] d\psi \leq \tau\}$. Then

$$\mathbb{P}_{\pi^\beta}[w_r^T \sum_{t=1}^{\infty} \gamma^{t-1} \phi(s_t) \geq Q^{\pi_\beta}(s, \hat{\pi}(s))]$$

$$= \mathbb{P}_{\psi \sim p_0(\cdot \mid s)}[w_r^T \psi \geq w_r^T \psi^*]$$

$$= \int 1_{[w_r^T \psi \geq w_r^T \psi^*)]} p_0(\psi \mid s) d\psi$$

$$\leq \int 1_{[w_r^T \psi \geq \widehat{R}]} p_0(\psi \mid s) d\psi$$

$$= \int 1_{[w_r^T \psi \geq \widehat{R}]} \hat{p}(\psi \mid s) d\psi$$

$$\quad + \int 1_{[w_r^T \psi \geq \widehat{R}]} \left[ p_0(\psi \mid s) - \hat{p}(\psi \mid s) \right] d\psi$$

$$\leq \tau + \int 1_{[w_r^T \psi \geq \widehat{R}]} \epsilon d\psi$$

$$\leq \tau + \epsilon.$$

$\square$

## A.2 Proof of Corollary 5.4

*Proof.* Intuitively, policy $\hat{\pi}$ fall behind by at most $\lambda(\tau + \epsilon)$ at each time step: $\forall s_1 \in S$,

$$
\begin{aligned}
& V^*(s_1) - V^{\hat{\pi}}(s_1) \\
&= Q^*(s_1, a^*) - Q^{\hat{\pi}}(s_1, \hat{a}) \\
&= Q^*(s_1, a^*) - Q^*(s_1, \hat{a}) \\
&\quad + Q^*(s_1, \hat{a}) - Q^{\hat{\pi}}(s_1, \hat{a}) \\
&\leq \lambda(\tau + \epsilon) + Q^*(s_1, \hat{a}) - Q^{\hat{\pi}}(s_1, \hat{a}) \\
&= \lambda(\tau + \epsilon) + \gamma \mathbb{E}_{s_2 \sim p(s_1, \hat{a})}[V^*(s_2) - V^{\hat{\pi}}(s_2)] \\
&\leq \lambda(\tau + \epsilon) + \gamma\lambda(\tau + \epsilon) \\
&\quad + \gamma^2 \mathbb{E}_{s_3}[V^*(s_3) - V^{\hat{\pi}}(s_3)] \\
&\leq \cdots \\
&\leq \sum_{i=0}^{\infty} \gamma(\tau + \epsilon) \\
&= \frac{\lambda}{1 - \gamma}(\tau + \epsilon).
\end{aligned}
$$

$\square$

## A.3 Proof of Theorem 5.5

*Proof.* Under the full coverage condition, the sampling optimality $\tau$ can be set to be zero for concrete actions, and the condition $p(\psi|s) > \epsilon$ becomes $p(\psi|s) > 0$. Then we see that planning only focuses on the supporting set of $\hat{p}$: $\hat{p}(\psi \mid s) = 0$ if and only $(s, \psi) \in \mathcal{D}$ for some time during execution, which equals $p_0(\psi \mid s) = 0$. This indicates that $\text{supp}(\hat{p}) = \text{supp}(p_0)$. Therefore the conclusion follows. $\square$

# B   Alternative Derivation of Guided Diffusion Sampling

In this section, we derive the guided diffusion sampling from the perspective of control as inference [33]. To start, we define the trajectory-level optimality variable $\mathcal{O}$ as a Bernoulli variable taking the value of 1 with probability $\exp(R(\tau))$ and 0 otherwise, where $R(\tau) = \sum_{t=0}^{T} \gamma^t r(s_t) - R_{\max}$. Note we subtract the max discounted return $R_{\max}$ to make the density a valid probability distribution. Planning can be cast as an inference problem where the goal is to sample $\psi^* \sim p(\psi|\mathcal{O})$. By Bayes rule, we have

$$ p(\psi|\mathcal{O}) \propto p(\mathcal{O}|\psi)p(\psi) $$

Taking the gradient of the log of both sides, we get

$$
\begin{aligned}
\nabla_\psi \log p(\psi|\mathcal{O}) &= \nabla_\psi \log p(\mathcal{O}|\psi) + \nabla_\psi \log p(\psi) \\
&= \nabla_\psi \log \exp(w^\top \psi) + \nabla_\psi \log p(\psi) \\
&= \nabla_\psi w^\top \psi + \nabla_\psi \log p(\psi) \\
&= w + \nabla_\psi \log p(\psi)
\end{aligned}
$$

This implies that we can sample from $p(\psi|\mathcal{O})$ by adding the regression weights $w$ to the score at each timestep, yielding the same guided diffusion form as in Sec. 4.2.

# C   Implementation Details

## C.1 Model architecture

We parameterize the random Fourier features using a randomly initialized 2-layer MLP with 2048 units in each hidden layer, followed by sine and cosine activations. For a $d$-dimensional feature,

the network's output dimension is $\lfloor d/2 \rfloor$ and the final feature is a concatenation of sine and cosine activated outputs. We set $d = 128$ for all of our experiments.

We implement the outcome model and policy using conditional DDIMs [47]. The noise prediction network is implemented as a 1-D Unet with down dimensions $[256, 512, 1024]$. Each layer is modulated using FiLM [41] to support conditioning.

## C.2 Training details

We train our models on the offline dataset for 100,000 gradient steps using the AdamW optimizer [35] with batch size 2048. The learning rate for the outcome model and the policy are set to $3e^{-4}$ and adjusted according to a cosine learning rate schedule with 500 warmup steps. We train the diffusion noise prediction network with 1000 diffusion timesteps and sample using the DDIM [47] sampler with 50 timesteps. We sample from an exponential moving average model with decay rate $0.995$. The same set of training hyperparameters is shared across all environments.

To transfer to a downstream task, we randomly sample 10000 transitions from the dataset, relabel them with the test time rewards, and perform linear least squares regression. We use the guided diffusion planner for Antmaze ($\alpha = 0.5$), Franka Kitchen ($\alpha = 0.01$), and Roboverse ($\alpha = 0.05$) experiments. For Hopper, we use the random shooting planner with 1000 particles. We found planning with guided diffusion to be sensitive to the guidance coefficient. **Hence, for new environments, we suggest using the random shooting planner to get a baseline performance and then tuning the guided diffusion coefficient to acclerate inference.** We run each experiment with 6 random seeds and report the mean and standard deviation in the tables. Each experiment (pretraining + adaptation) takes 3 hours on a single Nvidia L40 GPU.

## C.3 Baselines

**Successor Features**   We use the implementation of Universal Successor Features Approximators [5] and Forward-Backward Representation [54, 55] from the code release of [55]. We choose random Fourier features for universal SF as it performs best across the evaluation suite. To ensure fairness of comparison, we set the feature dimension to be 128 for both Universal SF and FB. Both methods are pretrained for 1 million gradient steps and adapted using the same reward estimation method as DiSPOs.

**RaMP**   We adapt the original RaMP implementation [9] and convert it into an offline method. RaMP originally consists of an offline training and an online adaptation stage, where online adaptation alternates between data collection and linear regression. We instead adapt by subsampling 10000 transitions from the the offline dataset and relabeling them with the test-time reward function, thus removing the exploration challenge. We use an MPC horizon of 15 for all experiments.

**Model-based RL**   We use the original implementations of MOPO [64] and COMBO [63] in our evaluations. We pretrain only the trainsition model on the offline transition datasets. To transfer to downstream tasks, we freeze the transition model, train the reward model on state reward pairs, and optimize the policy using model-based rollouts. We set the model rollout length for both methods to 5 and the CQL coefficient to be 0.5 for COMBO.

**Goal-conditioned RL**   We use the GC-IQL baseline from [40]. To remove the privileged information, we modify the sampling distribution to only sample from half of the goal space excluding the test-time goal location.

# D  Environment Details

**Antmaze**   [16] is a navigation domain that involves controlling an 8-DoF quadruped robot to reach some designated goal location in a maze. Each task corresponds to reaching a different goal location. We use the standard D4RL offline dataset for pretraining. For downstream task adaptation, we replace the standard sparse reward $\mathbb{1}(s = g)$ with a dense reward $\exp(-||s - g||_2^2/20)$ to mitigate the challenge of sparse reward in long-horizon problems.

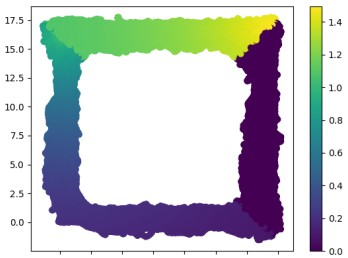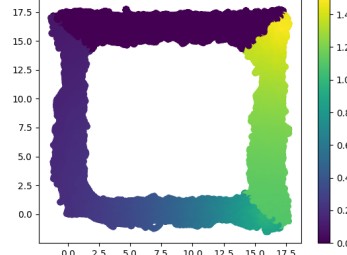

Figure 6: Data distribution and reward for Antmaze Preference environments. The left figure illustrates a preference for taking the vertical path, and the right figure illustrates the preference for taking the horizontal path.

**Franka Kitchen** [18] is a manipulation domain where the goal is to control a Franka arm to interact with appliances in the kitchen. Each task corresponds to interacting with a set of items in no particular order. We use the standard D4RL offline dataset for pretraining. For downstream task adaptation, we use the Markovian sparse rewards, where at each timestep the robot gets a reward equal to the number of completed tasks. We report the number of tasks completed throughout the entire episode in Table 1.

**Hopper** Hopper [7, 9] is a locomotion domain where the task is to control a hopper to perform various tasks, including hopping forward, hopping backward, standing, and jumping. We design a shaped reward for each task. We pretain on the offline dataset from [9], collected by taking the replay buffers of expert SAC [19] agents trained to solve each task.

**Preference Antmaze** is a variant of D4RL Antmaze [15] where the goal is to reach the top right cell from the bottom left cell in a custom maze shown in Fig. 4. The two tasks in this environment are the two paths to reaching the goal, simulating different human preferences. To construct the dataset, we collect 1 million transitions using the D4RL waypoint controller. For each preference, we design a reward function that encourages the agent to take one path and not the other. Fig. 6 visualizes the dataset and the reward function for each preference.

**Roboverse** [46] is a tabletop manipulation environment consisting of a WidowX arm aiming to complete multi-step problems. Each task consists of two phases, and the offline dataset contains separate trajectories for each phasebut not full task completion. We use the standard sparse reward, assigning a reward of 1 for each timestep the task is completed.

# E    Additional Experiments

To understand the impact of various design decisions on the performance of DiSPOs, we conducted systematic ablations on the various components, using the Antmaze medium-diverse as a test bed.

## E.1    Full D4RL Results

Table. 4 displays the full D4RL results with oracle goal-conditioned baseline labeled **GC-Oracle**. The baseline is trained on a goal distribution covering the testing-time goals, granting it privileged information. Despite this, DiSPOs are competitive with the oracle in most domains.

## E.2    Ablation of Planning Method

We compare the guided diffusion planner with the random shooting planner described in Sec. 4.2. As shown in Table 5, the guided diffusion planner achieves comparable performance to random shooting with 1000 particles while taking significantly less wall-clock time. While we can decrease the number of samples in the random shooting planner to improve planning speed, this comes at the cost of optimality.

Table 4: Full offline multitask RL on AntMaze and Kitchen. DiSPOs show superior transfer performance (in average episodic return) than successor features, model-based RL, and misspecified goal-conditioned baselines, while being competitive with an oracle using privileged information.

| | DiSPO (Ours) | USFA | FB | RaMP | MOPO | COMBO | GC-IQL | GC-Oracle |
|---|---|---|---|---|---|---|---|---|
| umaze | **593** $\pm$ 16 | 462 $\pm$ 4 | 469 $\pm$ 12 | 459 $\pm$ 3 | 451 $\pm$ 2 | 574 $\pm$ 10 | 571 $\pm$ 15 | 623 $\pm$ 7 |
| umaze-diverse | **568** $\pm$ 12 | 447 $\pm$ 3 | 474 $\pm$ 2 | 460 $\pm$ 7 | 467 $\pm$ 5 | 547 $\pm$ 11 | 577 $\pm$ 7 | 576 $\pm$ 43 |
| medium-diverse | **631** $\pm$ 67 | 394 $\pm$ 52 | 294 $\pm$ 61 | 266 $\pm$ 2 | 236 $\pm$ 4 | 418 $\pm$ 16 | 403 $\pm$ 10 | 659 $\pm$ 44 |
| medium-play | **624** $\pm$ 58 | 370 $\pm$ 31 | 264 $\pm$ 29 | 271 $\pm$ 5 | 232 $\pm$ 4 | 397 $\pm$ 12 | 390 $\pm$ 33 | 673 $\pm$ 45 |
| large-diverse | **359** $\pm$ 59 | 215 $\pm$ 20 | 181 $\pm$ 46 | 132 $\pm$ 1 | 128 $\pm$ 1 | 244 $\pm$ 19 | 226 $\pm$ 9 | 493 $\pm$ 9 |
| large-play | **306** $\pm$ 18 | 250 $\pm$ 41 | 165 $\pm$ 12 | 134 $\pm$ 3 | 128 $\pm$ 2 | 248 $\pm$ 4 | 229 $\pm$ 5 | 533 $\pm$ 8 |
| kitchen-partial | **43** $\pm$ 6 | 0 $\pm$ 0 | 4 $\pm$ 4 | 0 $\pm$ 0 | 8 $\pm$ 7 | 11 $\pm$ 9 | - | 33 $\pm$ 23 |
| kitchen-mixed | **46** $\pm$ 5 | 10 $\pm$ 10 | 5 $\pm$ 5 | 0 $\pm$ 0 | 0 $\pm$ 0 | 0 $\pm$ 0 | - | 43 $\pm$ 7 |
| hopper-forward | 566 $\pm$ 63 | 487 $\pm$ 110 | 452 $\pm$ 59 | 470 $\pm$ 16 | 493 $\pm$ 114 | **982** $\pm$ 157 | - | - |
| hopper-backward | 367 $\pm$ 15 | 261 $\pm$ 68 | 269 $\pm$ 77 | 220 $\pm$ 15 | **596** $\pm$ 211 | 194 $\pm$ 74 | - | - |
| hopper-stand | **800** $\pm$ 0 | 685 $\pm$ 130 | 670 $\pm$ 120 | 255 $\pm$ 15 | **800** $\pm$ 0 | 600 $\pm$ 111 | - | - |
| hopper-jump | **832** $\pm$ 22 | 746 $\pm$ 112 | 726 $\pm$ 35 | 652 $\pm$ 28 | 753 $\pm$ 51 | 670 $\pm$ 109 | - | - |

Table 5: Ablation of planning method. Wall time is measured over 1000 planning steps.

| | Return $\uparrow$ | Wall time (s) $\downarrow$ |
|---|---|---|
| DiSPO (Ours) | 631 $\pm$ 67 | 42.9 |
| Random shooting @ 1000 | 650 $\pm$ 50 | 94.8 |
| Random shooting @ 100 | 619 $\pm$ 90 | 58.6 |
| Random shooting @ 10 | 513 $\pm$ 52 | 55.5 |

### E.3 Ablation of Feature Dimension and Type

To understand the importance of feature dimension and type, we compare variants of our method that use lower-dimensional random Fourier features, vanilla random features, and the two top-performing pretrained features from [55]. From Table. 6 we observe that as feature dimension decreases, their expressivity diminishes, resulting in lower performance. We found random features to perform much worse than random Fourier features. Interestingly, pretrained features with dynamics prediction and graph Laplacian objectives also achieve lower returns than random Fourier features. We hypothesize these pretrained features overfit to the training objective and are less expressive than random Fourier features

### E.4 Ablation of Dataset Coverage

We investigate the effect of dataset coverage on the performance of our method. We compare DiSPOs trained on the full D4RL dataset against two variants, one where we randomly subsample half of the transitions, and the other where we adversarially remove the transitions from the half of the state space containing the test-time goal. As shown in Table 7, the performance of DiSPO drops as dataset coverage degrades.

### E.5 Ablation of Planning Horizon

In Fig. 8, we ablate the effective planning horizon of DiSPOs by controlling the discount factor $\gamma$. We found that for shorter horizon tasks (umaze-diverse), the optimality of planned trajectories improves as the planning horizon decreases. However, for long-horizon tasks (medium-diverse), reducing the planning horizon too much incurs cost on global optimality.

### E.6 Nonparametric Baseline

To confirm the intuition of our method, we implemented a nonparametric baseline that constructs an empirical estimate of the outcome distribution. First, we calculate the empirical discounted sum of

Table 6: Ablation of feature dimension and type.

|  | Return ↑ |
| --- | --- |
| DiSPO (Ours) | $631 \pm 67$ |
| Random Fourier (64-dim) | $561 \pm 45$ |
| Random Fourier (32-dim) | $295 \pm 30$ |
| Random Fourier (16-dim) | $307 \pm 38$ |
| Random | $382 \pm 43$ |
| Forward dynamics | $402 \pm 36$ |
| Laplacian | $376 \pm 33$ |

Table 7: Ablation of dataset coverage.

|  | Return ↑ |
| --- | --- |
| Full dataset | $631 \pm 67$ |
| Random Subsampling | $459 \pm 57$ |
| Adversarial Subsampling | $390 \pm 26$ |

Table 8: Ablation of planning horizon.

|  | umaze-diverse | medium-diverse |
| --- | --- | --- |
| $\gamma = 0.99$ | $587 \pm 12$ | $650 \pm 50$ |
| $\gamma = 0.95$ | $597 \pm 16$ | $\mathbf{682} \pm \mathbf{20}$ |
| $\gamma = 0.9$ | $\mathbf{601} \pm \mathbf{19}$ | $553 \pm 36$ |

Table 9: Comparison to a nonparametric baseline that takes the top-valued action among the $k$ nearest neighbors of a state.

|  | Return ↑ |
| --- | --- |
| DiSPO (Ours) | $631 \pm 67$ |
| Nonparametric $k = 10$ | $308 \pm 20$ |
| Nonparametric $k = 100$ | $299 \pm 21$ |
| Nonparametric $k = 1000$ | $287 \pm 10$ |

features along trajectories in the dataset. Given the set of empirical $(s, a, \psi)$ pairs and the downstream reward weight $w$, we can then select the optimal action at state $s$ through the following steps: (1) query the $k$ nearest neighbors of $s$, (2) evaluate their corresponding values $w^\top \psi$, (3) take the action of the top-valued neighbor.

We found this baseline to perform surprisingly well on antmaze-medium-diverse-v2. While it does not achieve the performance of DiSPOs, it outperforms RaMP and MOPO. This result confirms the intuition behind DiSPOs, which involves selecting the optimal outcome under dataset coverage and taking an action to realize it. We attribute the performance gap between this baseline and DiSPOs to their trajectory stitching ability (acquired via dynamic programming), infinite horizon modeling, and neural network generalization.

### E.7 Visualization of Stitching

We visualize the stitched trajectories for roboverse environments in Fig. 7. The dataset only contains trajectories from each phase separately, but DiSPOs can generate full trajectories by stitching the subtrajectories.

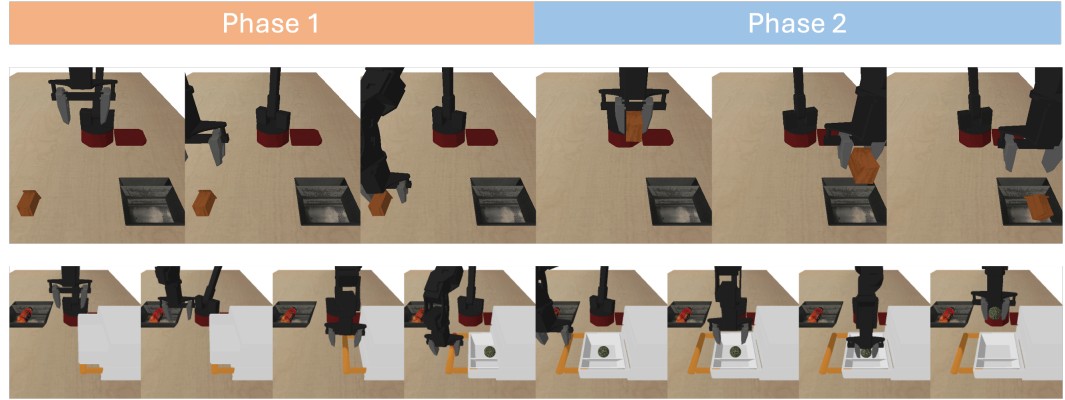

Figure 7: Visualization of stitched trajectories for roboverse PickPlace and BlockedDrawer.

# F    Algorithm Pseudocode

---

**Algorithm 1** DiSPO Training

---

1: Given transition dataset $\mathcal{D}$, feature function $\phi(\cdot)$
2: Initialize $p_\theta(\psi|s)$, $\pi_\rho(a|s,\psi)$.
3: **while** not converged **do**
4:     Draw $B$ transition tuples $\{s_i, a_i, s_i'\}_{i=1}^B \sim \mathcal{D}$.
5:     Sample successor features for next states $\psi_i' \sim p_\theta(\psi|s_i')$, $i = 1 \dots N$.
6:     Construct target successor feature $\psi_i^{\text{targ}} = \phi(s) + \gamma * \psi_i'$.
7:     // $\psi$ model learning
8:     Update feature distribution: $\theta \leftarrow \arg\max_\theta \log p_\theta(\psi_i^{\text{targ}}|s_i)$.
9:     // Policy extraction
10:     Update policy: $\rho \leftarrow \arg\max_\rho \log \pi_\rho(a|s_i, \psi_i^{\text{targ}})$.
11: **end while**

---

**Algorithm 2** DiSPO Offline Adaptation

---

1: Given transition dataset $\mathcal{D}$, feature function $\phi(\cdot)$, reward function $r(s)$.
2: Relabel offline dataset $\mathcal{D}$ with reward function.
3: Initialize regression weights $w$.
4: Fit $w$ to $\mathcal{D}$ using linear regression $w = \arg\min_w \mathbb{E}_\mathcal{D}[\| w^\top \phi(s) - r(s) \|_2^2]$.

---

**Algorithm 3** DiSPO Online Adaptation

---

1: Given $p_\theta(\psi|s)$, $\pi_\rho(a|s,\psi)$, feature function $\phi(\cdot)$.
2: Prefill online buffer $\mathcal{D}_{\text{buf}}$ with random exploration policy.
3: Initialize regression weights $w$.
4: **for** time steps $1 \dots T$ do **do**
5:     Fit $w$ using linear regression $w = \arg\min_w \mathbb{E}_{\mathcal{D}_{\text{buf}}}[\| w^\top \phi(s) - r(s) \|_2^2]$.
6:     Infer optimal $\psi^* = \arg\max_\psi w^\top \psi$   s.t.   $p_\theta(\psi|s) > \epsilon$.
7:     Sample optimal action $a^* \sim \pi(a|s, \psi^*)$.
8:     Execute action in the environment and add transition to buffer.
9: **end for**

---

**Algorithm 4** DiSPO Inference (Random Shooting)

---

1: Given $p_\theta(\psi|s)$, $\pi_\rho(a|s,\psi)$, regression weight $w$, current state $s$.
2: Sample $N$ outcomes $\{\psi_i\}_{i=1}^N \sim p_\theta(\psi|s)$.
3: Compute corresponding values $\{v_i\}_{i=1}^N$, where $v_i = w^\top \psi_i$.
4: Take optimal cumulant $\psi^* = \psi_i$, where $i = \arg\max_i \{v_i\}_{i=1}^N$.
5: Sample optimal action $a^* \sim \pi(a|s, \psi^*)$.

---

**Algorithm 5** DiSPO Inference (Guided diffusion)

---

1: Given diffusion model $p_\theta(\psi|s)$, $\pi_\rho(a|s,\psi)$, regression weight $w$, current state $s$, guidance coefficient $\beta$.
2: Initialize outcome $\psi_1$ from prior.
3: **for** diffusion timestep $t = 1...T$ **do**
4:     Compute noise at timestep $\epsilon = \epsilon_\theta(\psi_t, t, s)$.
5:     Update noise $\epsilon' = \epsilon - \beta\sqrt{1 - \bar{\alpha}_t}w$.
6:     Sample next timestep action $\psi_{t+1}$ using $\epsilon'$.
7: **end for**
8: Sample optimal action $a^* \sim \pi_\rho(a|s, \psi_T)$.

---

