# OpenReview forum: "Distributional Successor Features Enable Zero-Shot Policy Optimization"
_NeurIPS.cc/2024/Conference — NeurIPS 2024 poster_

### Official Review · Reviewer_EMA6 · 2024-07-10

**Soundness:** 2
**Presentation:** 3
**Contribution:** 3
**Rating:** 5
**Confidence:** 2

**Summary:**

The paper presents a novel approach called Generalized Occupancy Models (GOMs), which aims to address the challenges of model-based RL and successor features in transferring across tasks with various reward functions. GOMs learn a distribution of successor features from a stationary dataset, enabling the quick selection of optimal actions for new tasks without suffering from compounding error. The paper provides a theoretical analysis of GOMs and demonstrates their efficacy in simulated robotics problems.

**Strengths:**

This work is overall well written and presented. It provides an interesting theoretical analysis of the proposed method, which is complemented by a decent empirical validation through experiments in various simulated robotics problems, demonstrating the practical applicability of GOMs. The paper also provides details regarding implementation alongside the codebase to enable reproducibility.

**Weaknesses:**

In general, the paper appears to oversell the generalization capabilities a bit. One of the main assumptions is the shared state space across tasks, which is often simply not the case even when assuming the same agent/embodiment and closely related tasks (different number of objects/obstacles, different goal specifications, etc.).
This is to the best of my understanding also reflected to some extend in the experiments, which seem to leverage the same task, merely changing the goal or sequence. In the experiments I would like to see actual changes of the underlying reward function of performing a completely different task in the same setting. This could for example include stacking cubes vs moving them away from each other, etc.

Another concern is the statistical significance of the presented results. The experiments only report the mean over four seeds with 1 standard deviation, which might limit the statistical significance of the presented results. A more robust experimental setup with a larger number of seeds and a thorough analysis of the standard deviation would provide a more convincing argument for the efficacy of GOMs.

As acknowledged by the authors, GOMs require a pre-specified feature function that linearly expresses rewards, which may not always be feasible or accurate for all types of environments or tasks. However, the paper does not provide any further insights why Fourier feature perform better than others and if this is a general phenomenon with the proposed method or if this is something that needs to be tuned/selected for different tasks.


The paper may overstate its generalization capabilities. A significant assumption underpinning the framework is the consistency of the state space across tasks. In practical scenarios, even with the same agent or embodiment, state spaces can vary significantly, such as in the number of objects or obstacles present or in task goal specification. This limitation is, to the best of the reviewers knowledge, somewhat reflected in the experimental design, which seems to focus on variations of the same task by altering goals or sequences rather than fundamentally changing the underlying reward functions. The experiments could be more convincing if they included tasks that are distinctly different within the same environment, such as contrasting tasks of stacking cubes versus moving them apart.

The statistical robustness of the results is another area of concern. The experiments report averages over only four seeds with 1 standard deviation, which may not sufficiently demonstrate the reliability of the findings. A more comprehensive experimental setup with a greater number of seeds would strengthen the case for the effectiveness of GOMs.

While the authors recognize that GOMs depend on a pre-specified feature function capable of linearly expressing rewards, the paper only provides some analysis on why Fourier features were chosen and whether their performance is superior to other features in the appendix. This also raises the question whether Fourier features are generally applicable for the proposed method or if it is a parameter that needs to be fine-tuned or selected for different tasks/rewards. Addressing these points could further enhance the paper's contributions and provide clearer guidance on the applicability and limitations of GOMs.

**Questions:**

- Why is Franka Kitchen using sparse rewards while this was changed for antmaze?
- Is the guidance coefficient also sensitive to different rewards within the same environment or just to the environment itself?

**Limitations:**

The authors sufficiently discuss the limitations of their work.

---

> ### Author Rebuttal · Authors · 2024-08-07
>
> Thank you for your positive feedback and finding our approach “novel”. We address your questions below.
>
> > (W1) In the experiments I would like to see actual changes of the underlying reward function of performing a completely different task in the same setting.
>
> We conduct additional experiments on the Hopper environment adapted from RaMP. The environment features 4 tasks: forward, backward, stand, and jump, with complex reward functions that are not easily specified by goals. The dataset is a mixture of replay buffers of expert agents trained to achieve these tasks. We adapt the environment and dataset to our offline setting, relabeling the dataset with different task rewards for transfer. GOM compares favorably to representative baselines from each class of methods, achieving the highest average return across 4 tasks.
>
> > (W2) Another concern is the statistical significance of the presented results.
>
> We recognize the importance of having more seeds in RL experiments. Due to the compute bottleneck, we present results with 6 seeds for a subset of the main experiments in Table 6 of the attached pdf. While we found no significant variation in the results, potentially due to the offline nature of the algorithms, we will update the results in the final revision with 6 seeds.
>
> > (W3) However, the paper does not provide any further insights why Fourier feature perform better than others and if this is a general phenomenon with the proposed method or if this is something that needs to be tuned/selected for different tasks.
>
> For antmaze and kitchen, we found random Fourier features to outperform random features and learned  features. Recent work [1] shows that Fourier features can fit higher frequency functions, explaining its improved expressivity compared to standard random features. The reason it outperforms learned features is rather elusive, and we hypothesize that this is because the antmaze and kitchen rewards are rather structured. Hence it suffices to fit the reward with Fourier features, while learned features overfit to their pretraining objectives. For more complicated rewards in the Hopper tasks (Table 1 in pdf), we found a feature network pretrained with dynamics prediction objective to outperforms fourier feature. This applies to both our method and the USF baseline.
>
> > (Q1) Why is Franka Kitchen using sparse rewards while this was changed for antmaze?
>
> We use a dense reward for antmaze because the task is long horizon in nature, making it difficult to propagate sparse reward signals via dynamic programming. The kitchen tasks, on the other hand, have shorter horizons. Moreover, they use a stagewise sparse reward, assigning a reward equal to the number of completed stages (less sparse than a task-completion reward). This is a standard reward function for these types of multistage tasks [2].
>
> > (Q2) Is the guidance coefficient also sensitive to different rewards within the same environment or just to the environment itself?
>
> We found the same guidance coefficient to work well for all tasks within antmaze, kitchen, and hopper. Hence from our set of experiments the guidance coefficient is only sensitive to the environment itself but not the different rewards.
>
> References:
>
> [1] Ge Yang, Anurag Ajay, Pulkit Agrawal. Overcoming the Spectral Bias of Neural Value Approximation. ICLR 2022.
>
> [2] Justin Fu, Aviral Kumar, Ofir Nachum, George Tucker, Sergey Levine. D4RL: Datasets for Deep Data-Driven Reinforcement Learning. ArXiv 2020.

---

> ### Comment · Reviewer_EMA6 · 2024-08-12
>
> Thank you for your detailed response and for conducting the additional experiments. While I acknowledge the improvements and the potential benefits highlighted, I fully agree with reviewer tiLN's comment on the limited sample size and therefore I will not increase my score at this time.

---

> > ### Author Response · Authors · 2024-08-13
> >
> > We understand your concern and have updated the main results for GOM and Universal SF with 10 seeds. As shown in the following table, we don’t see a significant difference in terms of the mean and the standard deviation of the runs compared to Table 1. We hope this can alleviate your concern about our results, and we will be sure to run 10 seeds for all experiments in the revision.
> > | Task                | GOM         | USF         |
> > |---------------------|-------------|-------------|
> > | umaze-v2            | 591 ± 15    | 458 ± 6     |
> > | umaze-play-v2       | 576 ± 14    | 443 ± 5     |
> > | medium-diverse-v2   | 648 ± 54    | 373 ± 43    |
> > | medium-play-v2      | 625 ± 52    | 390 ± 34    |
> > | large-diverse-v2    | 345 ± 50    | 233 ± 28    |
> > | large-play-v2       | 327 ± 67    | 237 ± 39    |
> > | kitchen-partial     | 40 ± 7      |   1 ± 1     |
> > | kitchen-mixed       | 46 ± 9      |  9 ± 9       |

---

### Official Review · Reviewer_tiLN · 2024-07-13

**Soundness:** 3
**Presentation:** 3
**Contribution:** 3
**Rating:** 7
**Confidence:** 4

**Summary:**

The paper describes a method for using distributions of successor features for fast transfer across reinforcement learning tasks. The method combines the long-term predictions and fast transfer properties of successor features with a "readout policy" (conditioned on achieving a particular successor feature) and a way to sample good successor features for the readout policy to achieve (using diffusion models).

**Strengths:**

By modeling distributions over successor features, the paper argues that the model can avoid the policy-dependence of typical successor feature approaches. This is an important problem, and the paper does a good job of motivating the work.

The discussion of the model and its various components is clear.

The theoretical analysis is nice. While the theorems do not quite guarantee that the method will work well in practice, they are a good sign that the method should work under ideal conditions.

The experimental results are encouraging.

**Weaknesses:**

1. The paper's main argument in favor of the proposed method is that it overcomes the policy dependence of successor features, since the method is trained from a behavior dataset, rather than a particular policy.

I find this argument unconvincing. I understand that the behavior policy can be non-stationary, but the empirical distribution of the dataset itself implicitly defines a single policy. The paper could do a better job explaining how this is providing more diversity of experience than typical policy-dependent successor feature approaches.

If I had to guess, I'd say that path dependence in the trajectory/policy interactions means that structured policies generate more interesting trajectories that are hard to sample from a simple stochastic policy. For example, walking in a specific direction for a long time, versus taking a random walk. If we sample from the implicit policy defined by the empirical data distribution, we would get less structured behavior.

2. It's a little unclear to me which parts of the proposed method are new, and which are not. It's also unclear to me what the baseline successor feature methods would do in various places. For example, in the last paragraph of Section 4.2, maybe the paper could discuss what would normally be required to get S.F.s to work, so that it is clearer why GOMs represent a new class of models.

3. In Def 5.2, it is unclear where $s_t$ comes from in the return. Is it sampled from $\pi_\beta$? Or from $\pi$? And it's not immediately clear what $\pi$ is being used for here. It would be helpful if the paper gave an overview of why we need $\pi$ before introducing the definition.

One related concern here. This says that a single (s,a) pair is good if it gets lower return with low probability. But the probability that the policy gets high return would be a product over all the state-action pairs, right? Does that require a very strict $\delta$?

4. The experimental results only use 4 seeds, so it's hard to say if this will generalize. See Henderson et al. (2017, https://arxiv.org/abs/1709.06560) for a discussion on why this is so important.

5. It would be helpful to see an example of the trajectories that are being stitched together.

**Questions:**

No specific questions.

---

> ### Author Rebuttal · Authors · 2024-08-07
>
> Thank you for the positive feedback! It is encouraging to hear that our paper is addressing “an important problem.” We reply to your questions below.
>
>  > (W1) The paper could do a better job explaining how this is providing more diversity of experience than typical policy-dependent successor feature approaches.
>
> The offline dataset indeed defines an implicit policy. Compared to a policy trained to optimize a specific reward function, the implicit policy has less structure and more coverage. Carrying over your analogy, a task-optimal policy could be walking in a particular direction for a long time, whereas the implicit dataset policy could be taking a random walk. Importantly, we can extract behaviors for walking towards many different directions from a dataset of random walks, just by taking a subset of actions at each step. In contrast, the action distribution of the task-optimal policy at each step is already narrow, so if the target task requires taking a step towards a different direction, the policy contains no information about it. This explains how modeling the distribution of successor features in the dataset enables transfer to various rewards.
>
> > (W2) It's a little unclear to me which parts of the proposed method are new, and which are not.
>
> Our work is built on a line of work in successor features and distributional RL. SF [1] first demonstrates the feasibility of estimating successor features using Bellman backup and evaluating a given policy under new rewards using the linearity argument. RaMP [2] propose random features as a viable choice of features for SF and use open-loop planning to avoid policy dependency. Our main contributions are (1) demonstrating the feasibility of using diffusion models to learn a distributional successor feature given a dataset, and (2) the ability to extra optimal policies for arbitrary rewards via a readout policy and guided diffusion planning. We will add this clarification to the related work section of the paper.
>
> > (W3) In Def 5.2, it is unclear where 𝑠𝑡 comes from in the return.
>
> $ \mathbb P_{\pi_\beta}[Q^{\pi_\beta}(s,a)< \sum_{t=1}^\infty \gamma^{t-1}r(s_t)\mid s_1=s]$ indicates that $s_t$ is from trajectories generated by following the dataset actions $\pi_\beta$ starting from state $s$. We will add this clarification in the revision.
>
> > (W4) The experimental results only use 4 seeds, so it's hard to say if this will generalize.
>
> We recognize the importance of having more seeds in RL experiments. Due to the compute bottleneck, we present results with 6 seeds for a subset of the main experiments in Table 6 of the attached pdf. While we found no significant variation in the results, we will update the results in the final revision with 6 seeds.
>
> > (W5) It would be helpful to see an example of the trajectories that are being stitched together.
>
> In Fig 2 of the attached pdf, we visualize rollouts of the GOM policy on the roboverse PickPlace and BlockedDrawer tasks. The dataset only contains trajectories of the first phase (e.g. picking) or the second phase (e.g. placing). The GOM policy is able to generate a single trajectory that completes two phases.
>
> References:
>
> [1] André Barreto, Will Dabney, Rémi Munos, Jonathan J. Hunt, Tom Schaul, Hado van Hasselt, David Silver. Successor Features for Transfer in Reinforcement Learning. NeurIPS 2017.
>
> [2] Boyuan Chen, Chuning Zhu, Pulkit Agrawal, Kaiqing Zhang, Abhishek Gupta. RaMP: Self-Supervised Reinforcement Learning that Transfers using Random Features. NeurIPS 2023.

---

> > ### Comment · Reviewer_tiLN · 2024-08-09
> >
> > Thanks for the responses and for running the additional experiments. I think the benefits of accepting outweigh the risks, especially in light of the new experiments, but I won't increase my score due to the (still very low) number of seeds.
> >
> > Using 6 seeds is hardly any better than 4. I strongly urge you to ensure that these results hold for at least 10 seeds, and ideally 20-30. It will only strengthen the paper. Multiple reviewers made this point, and your future readers (who you also have to convince) will have the same skepticism.
> >
> > If compute is a bottleneck, that's an argument for running different experiments that are less computationally expensive---not an argument for lowering the standard of what counts as evidence. It is your responsibility to make your argument convincing, and 4-6 seeds are not convincing.

---

> > > ### Author Response · Authors · 2024-08-13
> > >
> > > We understand your concern and have updated the main results for GOM and Universal SF with 10 seeds. As shown in the following table, we don’t see a significant difference in terms of the mean and the standard deviation of the runs compared to Table 1. We hope this can alleviate your concern about our results, and we will be sure to run 10 seeds for all experiments in the revision.
> > > | Task                | GOM         | USF         |
> > > |---------------------|-------------|-------------|
> > > | umaze-v2            | 591 ± 15    | 458 ± 6     |
> > > | umaze-play-v2       | 576 ± 14    | 443 ± 5     |
> > > | medium-diverse-v2   | 648 ± 54    | 373 ± 43    |
> > > | medium-play-v2      | 625 ± 52    | 390 ± 34    |
> > > | large-diverse-v2    | 345 ± 50    | 233 ± 28    |
> > > | large-play-v2       | 327 ± 67    | 237 ± 39    |
> > > | kitchen-partial     | 40 ± 7      |   1 ± 1     |
> > > | kitchen-mixed       | 46 ± 9      |  9 ± 9       |

---

> > > > ### Comment · Reviewer_tiLN · 2024-08-13
> > > >
> > > > Thank you for running these additional seeds. I know this can be a hassle, but I appreciate that you did it anyway, and that you have promised to do the same for the remaining experiments. I will increase my score.

---

### Official Review · Reviewer_AWqu · 2024-07-15

**Soundness:** 4
**Presentation:** 3
**Contribution:** 4
**Rating:** 8
**Confidence:** 4

**Summary:**

This paper proposes an approach to zero-shot reinforcement learning through learning the distribution of successor features in deterministic MDPs allowing for the efficient computation of approximately optimal policies. The authors perform an empirical evaluation comparing their method to other zero-shot RL, model-based RL, and goal-conditioned RL methods showing gains in various continuous control domains.

**Strengths:**

- The way the authors learn a distribution over successor features is novel, i.e., using a maximum likelihood approach versus prior works that use MMD [1, 2].
- Using a diffusion model as a generative model of features through a loss employing bootstrapping is unique to this work as well as its application for planning.
- The paper is generally well-written and easy to read.


[1] Pushi Zhang, Xiaoyu Chen, Li Zhao, Wei Xiong, Tao Qin, and Tie-Yan Liu. Distributional Reinforcement Learning for Multi-Dimensional Reward Functions. Neural Information Processing Systems (NeurIPS), 2021.

[2] Harley Wiltzer, Jesse Farebrother, Arthur Gretton, Yunhao Tang, André Barreto, Will Dabney, Marc G. Bellemare, and Mark Rowland. A Distributional Analogue to the Successor Representation. International Conference on Machine Learning (ICML), 2024.

**Weaknesses:**

I have some serious issues with the framing of this paper. There are many contradictory statements and I believe the paper is a useful contribution but suffers from poor presentation and misleading claims. I'll outline my major concerns:

- I don't think the term generalized occupancy model is appropriate here. First, this isn't an occupancy model, we're modeling successor features not state occupancy as in [1]. Secondly, I don't believe the model is general in any sense of the word. As I'll discuss the method is policy dependent and is currently limited to deterministic MDPs.
- The paper gives a very confusing and somewhat misleading characterization of the policy-dependent nature of their method. There are many instances of this and I'll try to outline some of them:
	- In the introduction, "Rather than modeling the successor features under a particular policy, GOMs model the entire distribution of successor features under the behavior policy, ...", isn't the behavior policy not a particular policy?
	- "Importantly, DSM models the distributional successor measure of a particular policy, where the stochasticity stems purely from the policy and the dynamics. This makes it suitable for robust policy evaluation but not for transferring to arbitrary downstream tasks", If the DSM has these limitations and GOMs don't could you explain the difference between modeling the DSM over the mixture policy induced by a dataset when you have deterministic dynamics? If the answer is there's no difference then I don't think the framing of the DSM vs GOMs is a fair characterization.
	- "A key limitation of successor features is their inherent dependence on a single policy, as they are defined as the accumulated features when acting according to a particular policy. This makes extracting optimal policies for new tasks challenging.", again, given the problem setting you describe (i.e., fixed dataset, deterministic dynamics) I fail to see how this is a limitation. Learning SFs on this dataset have the same policy dependence as GOMs.
- The proposed method is closely related to 𝛾-models [2] and geometric horizon models [3]. Under deterministic dynamics, there's not much algorithmic novelty as GOMs reduce to the same cross-entropy TD objective in [2, 3] trained with a diffusion model.
- I believe the experimental methodology has some room for improvement, especially around the comparison to USFA and FB. See my questions below.

[1] Harley Wiltzer, Jesse Farebrother, Arthur Gretton, Yunhao Tang, André Barreto, Will Dabney, Marc G. Bellemare, Mark Rowland. A Distributional Analogue to the Successor Representation. CoRR abs/2402.08530, (2024)

[2] Michael Janner, Igor Mordatch, and Sergey Levine. Gamma-Models: Generative Temporal Difference Learning for Infinite-Horizon Prediction. Neural Information Processing Systems (NeurIPS), 2020.

[3] Shantanu Thakoor, Mark Rowland, Diana Borsa, Will Dabney, Rémi Munos, and André Barreto. Generalised Policy Improvement with Geometric Policy Composition. International Conference on Machine Learning (ICML), 2022.

**Questions:**

- You claim that random Fourier features performed best for USFA. Can you provide results for this? I would expect to see a comparison with either the Laplacian features from [1] or the HILP features from [2]. In fact, [2] shows that random features (not fourier) perform poorly compared to better base features for USFA.
- You chose to set the embedding dimensionality of FB to 128, did you sweep over this value? The embedding dimensionality has a regularizing effect in FB and in many tasks needs to be decreased to obtain good performance.
- Section 6.4, GOMs aren't the only method that can solve tasks with arbitrary rewards, why wasn't a comparison with USFA / FB performed here?
- How is the objective function (1) an off-policy update? This is a textbook example of an on-policy update, the target policy is the behavior policy (mixture policy over the dataset) by definition here.

[1] Ahmed Touati, Jérémy Rapin, and Yann Ollivier. Does Zero-Shot Reinforcement Learning Exist? International Conference on Learning Representations (ICLR), 2023.

[2] Park Seohong, Tobias Kreiman, and Sergey Levine. Foundation Policies with Hilbert Representations. International Conference on Machine Learning (ICML), 2024.

**Limitations:**

I don't believe the authors adequately addressed the limitations of their work. I believe assuming deterministic MDPs is too restrictive in practice. It's unclear how their method performs under any form of environment stochasticity which other methods like USFA, FB, and DSM do account for. At minimum, it would be nice for the authors to discuss this limitation in greater detail providing insight for how we can move beyond this limitation.

---

> ### Author Rebuttal · Authors · 2024-08-07
>
> Thank you for your constructive comments and for finding our paper “novel” and “well-written”. We address each point below.
>
> > (W1) I don't think the term generalized occupancy model is appropriate here.
>
> We name our method  “generalized occupancy model” because successor features capture a notion of state occupancy. Specifically, the successor measure is defined as $M^\pi(s_0, s) := \sum_t \gamma^t p(s_t = s| s_0, \pi)$, i.e. the discounted sum of probability of reaching state $s$ at each timestep. On the other hand, the state occupancy measure is defined as $\rho^\pi(s) := \mathbb E_{s_0 \sim p_0} \sum_t \gamma^tp(s_t=s | s_0, \pi)$. Therefore, the successor measure is the state occupancy measure with initial state distribution set to $\delta(s_0)$. Since successor feature is the integration of the feature function under the successor measure, it represents the expected feature under a state occupancy measure. That said, we realize the connection is not obvious and are open to changing the title to e.g. “Transferable Reinforcement Learning via Distributional Successor Features.” or another title that you feel might be more appropriate.
>
> > (W2.1, W2.3) In the introduction ... isn't the behavior policy not a particular policy? … Learning SFs on this dataset have the same policy dependence as GOMs.
>
> By a “particular policy” we mean a policy that is trained to optimize a particular reward. Computing the successor feature allows one to quickly estimate its value under new rewards given the linear reward weights. However, standard SF does not provide a means to obtain an optimal policy for a new reward. That is to say, for a new reward function, we can evaluate how suboptimal the policy is, but we cannot make it better for this reward without retraining. What GOM does is we assume access to a dataset $D$, (which can indeed be represented by a behavior policy), and for any reward $r$, we can find the best policy for this reward within the support of the dataset. So while GOM cannot go beyond the behavior dataset, it is not restricted to evaluating the mean return of behavior policy. Instead, it can recover the best policy contained within the support of $D$ for arbitrary reward.
>
> > (W2.2) ... could you explain the difference between modeling the DSM over the mixture policy induced by a dataset when you have deterministic dynamics?
>
> We realize that the characterization of DSM as having these limitations while GOMs don't is indeed a bit misleading. Conceptually, GOM is equivalent to DSM applied to a mixture policy induced by a dataset under the assumption of deterministic dynamics. *The key distinction between these two work is that our work shows DSM combined with deterministic dynamics can be used for transfer across rewards.* This insight is not shown conceptually or experimentally within the DSM paper. Rather, the DSM paper is motivated by robust zero-shot policy evaluation and risk-sensitive policy selection, a valuable contribution in a different light. *In the revision, we will position GOM as exploring new capabilities of a DSM-style framework, instead of addressing the limitations of DSM.*
>
> > (W3) The proposed method is closely related to 𝛾-models [2] and geometric horizon models [3].
>
> While both GOMs and $\gamma$-models / geometric horizon models learn the discounted occupancy measure under the dynamics, they differ in their transferability. Gamma models model the geometrically discounted state distribution of a policy trained to optimize a specific reward. While we can draw samples from this distribution to evaluate this policy under new rewards, we cannot improve the policy without re-optimization. On the other hand, GOMs learn a distribution over outcomes in the dataset, which can then be used to extract optimal policies for downstream tasks as long as it is covered by the data distribution.
>
> > (Q1) Ablation of feature type for USFA
>
> We provide additional ablation studies of the feature choice for USFA in Table 2 of the attached pdf. On the antmaze-medium task, we found Fourier features to outperform transition, laplacian, and HILP features. We hypothesize that the periodic structure in Fourier features is suitable for structured rewards such as distance-to-goal in antmaze.
>
> > (Q2) Sweep over embedding dimensionality of FB
>
> We provide additional ablations over the feature dimension of FB representation in Table 3 of the attached pdf. We found decreasing the feature dimension to provide a regularization effect, although decreasing it too much restricts the representation capacity.
>
> > (Q3) Section 6.4, why wasn't a comparison with USFA / FB performed here?
>
> We omitted these baselines because we were not trying to compare GOMs to USFA / FB on this specific task. Rather, we were trying to demonstrate GOM’s ability to optimize arbitrary rewards, which goal-conditioned methods cannot. We add the comparisons to USFA/FB in Table 4 of the pdf. As expected, USFA / FB can distinguish the different human preferences by observing the rewards, though they perform slightly worse than GOM.
>
> > (Q4) How is the objective function (1) an off-policy update?
>
> We say this is an off-policy update in the sense that we can use data from some dataset / behavior policy to find an optimal policy for a different task. That said, we understand your point about it being an on-policy update on the behavior policy, and we will add a clarification accordingly.
>
> > (Limitations) It's unclear how their method performs under any form of environment stochasticity.
>
> While our theoretical derivations are conducted with the deterministic MDP assumption, in practice the environments we evaluate on do contain stochasticity (lines 356-358), and we find our method to perform well. To address the deterministic assumption, we need to separately account for the stochasticity of the environment and the policy. We will add the limitation of deterministic dynamics assumption to the conclusion section of the revision.

---

> ### Comment · Reviewer_AWqu · 2024-08-13
>
> I thank the authors for their rebuttal. Significant effort has been put into the rebuttal and with the proposed modifications I believe this paper can be a strong contribution to a burgeoning field that studies what I'll call zero-shot policy optimization. I respond to the author(s) rebuttal below but I want to summarize the primary modification required for me to fight for this paper: remove the term generalized occupancy model. As I outline below it's confusing for multiple reasons and doesn't highlight the unique aspect of this work: the ability to perform zero-shot policy optimization in deterministic MDPs by modeling the distributional successor measure or a more computationally tractable objective in modeling distributional successor features.
>
> I want to raise my score to an 8 based on the proposed modifications making it to the final version of the paper as well as the additional empirical results exploring different base features and a more thorough investigation of the FB baseline. I strongly believe with these modifications this paper makes for a strong contribution to the community.
>
> ---
>
> (W1): It's not that I didn't understand the connection, it's that I believe the current naming is misleading if we are modeling features instead of states, especially in light of the distributional successor measure. I would like to see the title and naming changed for various reasons (more described below). On "Transferable Reinforcement Learning via Distributional Successor Features." I think this still undersells the work, I like the emphasis on distributional successor features but I think emphasizing that you're performing "zero-shot policy optimization" is important. When I see this title I immediately think of the title of the SF paper in which case transfer can mean policy evaluation instead of policy optimization.
>
> (W2.1, W2.3): Thanks for this explanation, I think I'm being pedantic concerning naming at this point, when I see occupancy the first question in my mind is: occupancy of what? I'm immediately thinking about a policy. I STRONGLY suggest changing the naming of the method, not only because of my point above but to highlight what's unique here. The fact your method can do "zero-shot policy optimization" is what's important so having planning or policy in the name seems important to help clear up confusion.
>
> (W2.2): I really like the idea of discussing your method as a new capability of the DSM, I think this will help clear up some confusion and better highlight the contributions of this work.
>
> (W3): I feel like we took one step forward and now one step back. I don't like this framing of GOM vs. gamma models / GHMs. Again, the unique thing about this work is the ability to perform zero-shot policy optimization under deterministic dynamics and how your specific choice of a GHM leads to an efficient planning routine. The cross-entropy TD loss in the GHM paper is essentially identical to the generative modeling component in your work, the only thing that differs is the model and that you're modeling features instead of states.
>
> (Q1, Q2, Q3): I thank the authors for these additional experiments, they helped solidify the strong performance of your method.

---

> > ### Author Response · Authors · 2024-08-13
> >
> > Thank you for recognizing our contribution to the field, and for engaging with us to help make the work properly scoped and positioned! Emphasizing "zero-shot policy optimization" in the title makes a lot of sense and will help distinguish our paper from related work. As a potential renaming, we propose changing the title to "Zero-Shot Policy Optimization via Distributional Successor Features" and naming our method "Diffusion Distributional Successor Features (DDSF)." We also agree on the comparison with GHM, and will emphasize that our key contribution is the ability to do zero-shot policy optimization. We will be sure to incorporate the additional experiments and discussions to the final revision.

---

### Official Review · Reviewer_7fQM · 2024-07-27

**Soundness:** 2
**Presentation:** 3
**Contribution:** 2
**Rating:** 4
**Confidence:** 3

**Summary:**

The paper proposes a method based on successor features for modeling possible long-term outcomes in the environment based on data, together with a learned policy to achieve those outcomes. After training from offline data, the model can produce a policy for a given new reward function without any additional interaction with the environment.

**Strengths:**

The general idea seems promising. Modeling long-term outcomes rather than single-step dynamics, like in regular model-based approaches, has the potential to help with the problem of error accumulation. This seems to be reflected in the evaluations performed.

The approach seems to outperform RaMP (although a modified version, made to be fully offline), another successor-feature-based method aiming to solve a similar issue of not being tied to a specific policy and being able to efficiently find policies optimizing for novel rewards.

**Weaknesses:**

1. The approach is explicitly modeling the distribution of outcomes in the training data. While it should have the ability to reach known outcomes from different initial configurations by performing trajectory stitching, it is unclear whether it could achieve any outcome not explicitly seen in the training data, even if it could be a simple combination of known ones. In contrast, model-based methods have the potential to allow for this, by reaching novel states by rolling out learned local models on novel action trajectories.

2. Another concern is related to finding the outcome that maximizes the given reward. Sampling in the space of all possible outcomes will not be scalable to more complicated environments, therefore it seems critical for the proposed conditional diffusion model to be able to find the desired outcome efficiently. It is difficult to judge based on the presented results that this is the case. In the ablation comparing the proposed method with random shooting, random shooting with 1000 samples outperforms the proposed method slightly while only taking twice as long to execute. It seems that for effectively finding the desired outcome in more complex environments this ratio would need to be much bigger, as we know that sampling will not be close to efficient enough.

3. The approach assumes access to the reward function in a functional form (or at least the value of it across the entire training dataset). It is unclear how realistic this assumption is. Other similar methods (like RaMP) use only the value of the new reward gotten from online interactions with the environment at test time.

The evaluations present might be insufficient to fully judge the quality of the approach:

4. Goal-Conditioned RL is trained on goals from only one half of the state space, while the proposed method is given data from the entire state space. This seems like inherently an unfair comparison. It would only be fair to give the same distribution to the proposed method for comparison.

5. Both the non-goal-conditioned evaluation and the trajectory stitching evaluation show simply that the method is capable of something in a most basic setup, beating only methods that by design are not capable of doing what is being evaluated. There should be more extensive evaluations to test how well they perform each of these functions, compared with other methods that might be plausibly used in such a case.

**Questions:**

Can a model produce any behavior that cannot be done by explicit stitching of trajectory chunks from the training data? Does one of the existing evaluations showcase this?

Is there a way, based on current experimental results, to reason about the efficiency of finding the desired outcome as the domain becomes more and more complex? Is this part expected to become the bottleneck in those cases?

**Limitations:**

It would be good to have a more clear comparison of pros on cons compared to baseline approaches the method is tested against, in particular related to some of the points raised above.

Additional evaluations would be valuable to be able to better judge the performance of the approach. Possibly, some of the evaluation environments from the RaMP paper could be used. Even with the proposed approach being fully offline it would be interesting to see how long it takes for methods evaluated there to catch up or exceed its performance. Most of the relevant baselines are already present in those evaluations -- the authors could possibly just run their approach and report those results.

---

> ### Author Rebuttal · Authors · 2024-08-07
>
> Thank you for your valuable feedback and for finding our approach “promising.” We address your comments below.
>
> > (W1) Model-based RL can generate novel action trajectories whereas GOM is constrained to the dataset distribution.
>
> While model-based RL can generate novel trajectories when queried with out-of-distribution actions, these trajectories are not guaranteed to be accurate since the model has never seen these transitions during training. Optimizing the policy under hallucinated trajectories can lead to model exploitation, resulting in suboptimal behavior when the policy goes out-of-distribution. In general, the best we can do in offline RL (either model-based or model-free) is to find the best in-distribution trajectory [1]. There can be in-distribution interpolation to new states, but not extrapolation to out-of-distribution states. In this sense, GOM and model-based RL have equal capabilities.
>
> > (W2, Q2) Concern about planning efficiency.
>
> Planning efficiency is a common concern shared by model-based planning methods, as one needs to search over actions over a long horizon in each planning step. In practice, planning with models is still reasonable because we typically don’t have to plan over the entire task horizon, just a shorter horizon (e.g. by dropping the discount factor) followed by replanning (as in model predictive control). This type of MPC reduces planning complexity while maintaining overall performance. Empirically, we experiment with planning over shorter horizons by decreasing the discount factor. As shown in Table 4 in the attached pdf, for short-horizon tasks such as antmaze-umaze, reducing the planning horizon indeed leads to better results when fixing the planning budget. On the other hand, for longer-horizon problems such as antmaze-medium, as one would expect, reducing planning horizon comes at the cost of global optimality.
>
> > (W3) The approach assumes access to the reward function in a functional form (or at least the value of it across the entire training dataset). It is unclear how realistic this assumption is.
>
> Our method does not inherently require access to the reward function in a functional form. Rather, it only requires a dataset of $(s, r)$ pairs to perform reward regression. This dataset can be collected using an exploration policy, or using the current task policy in an online setting, or by relabeling existing datasets when we have functional rewards. In particular, the reward relabeling scheme described in the paper is equivalent to rolling out the behavior policy to collect a new dataset for each new reward.
>
> > (W4) Goal-conditioned RL unfair comparison.
>
> We note that hindsight goal-conditioned RL consists of two distributions: the data distribution and the goal relabelling distribution. In our experiments, we remove the test-time goal from the goal-relabeling distribution, so the agent is still trained on all the transition data, but does not see the test-time goal. That said, we acknowledge that the comparison with goal-conditioned RL is rather delicate. With full goal converge, GCRL will be tested on training data, an unfair advantage compared to GOM which does not see test time reward function. On the other hand, covering goals renders GCRL incapable of learning behaviors to reach the goal. We provide results with both misspecified goal distribution and full goal distribution in Table 1 and 4 (Appendix) respectively to give a complete picture. We further emphasize that goal-conditioned RL is not generally applicable to the family of tasks we are considering, as shown in Table 5 of the attached pdf, while GOMs are naturally applicable to any task with Markovian rewards.
>
> > (W5) Non-goal-conditioned and trajectory stitching experiments show capability but not better performance to comparable methods.
>
> The experiments in Sections 6.4 and 6.5 are designed for analytical purposes to highlight the capability of GOMs. We do provide more experiments comparing the *performance* of GOM in terms of each capability. Specifically,  the kitchen-mixed environment in Table 3 of the paper requires explicitly stitching trajectories of completing different subtasks, as the dataset does not contain full task trajectory. We see that GOM outperforms baselines with trajectory stitching capability. To compare the performance of GOM on challenging non-goal-reaching tasks, we conduct additional experiments on the Hopper environment adapted from RAMP, where the agent is tasked with moving forward, moving backwards, standing, or jumping. As shown in Table 1, our method demonstrates competitive performance compared to the baselines, achieving the highest average return across 4 tasks.
>
> > (Q1) Can a model produce any behavior that cannot be done by explicit stitching of trajectory chunks from the training data? Does one of the existing evaluations showcase this?
>
> Stitching is typically achieved by dynamic programming (e.g. Bellman backup). GOM enables stitching via distributional Bellman backup, and hence exhibits the same stitching behavior as standard offline RL methods such as CQL, with the additional capability of transferring to new rewards.
>
> > (Limitations) Additional evaluations would be valuable to be able to better judge the performance of the approach. Possibly, some of the evaluation environments from the RaMP paper could be used.
>
> Thanks for suggesting additional evaluation environments. We adapt the Hopper environment from RaMP to the offline setting and compare our methods to representative baselines. We found our method to be competitive, achieving the highest average return across 4 tasks.
>
> References:
>
> [1] Sergey Levine, Aviral Kumar, George Tucker, Justin Fu. Offline Reinforcement Learning: Tutorial, Review, and Perspectives on Open Problems. ArXiv 2020.

---

> > ### Comment · Reviewer_7fQM · 2024-08-13
> >
> > > (W1)
> >
> > I am not sure I would fully agree with that characterization. For example, from MOPO, one of the model-based baselines the approach is compared with:
> >
> >     "For the algorithm to perform reliably, it’s crucial to balance the return and risk: 1. the potential gain in performance by escaping the behavioral distribution and finding a better policy, and 2. the risk of overfitting to the errors of the dynamics at regions far away from the behavioral distribution."
> >
> > Part of the evaluations in that paper is also focusing on "generalization to out-of-distribution behaviors".
> >
> > > (W2, Q2)
> >
> > My main focus was on how much better the proposed guided diffusion procedure is than simple random shooting. To add to my previous comment, in Table 5 in the supplementary material the performance of the proposed method and random shooting @ 100 are within each other's standard deviations (631 +/- 67 vs. 619 +/- 90). The proposed method is slightly faster (42.9s vs 58.6s), but random shooting also seems to have a significant overhead (55.5s for 10 vs 58.6s for 100) and it is unclear where it comes from.
> >
> > Additionally, one of the main selling points of the approach and differentiator from model-based approaches is the ability to model long-term outcomes. If the approach is instead used in an MPC fashion with a shorter horizon, differentiation from model-based methods becomes less clear as error accumulation might be less of an issue with a shorter horizon. It would be good to do a more extensive analysis of the proposed approach and model-based methods, both used in an MPC fashion, for a range of different planning horizons.
> >
> > > (W3)
> >
> > The main issue is the amount of (s, r) pairs needed for the new reward. While the behavior policy could be used to collect this data in an online fashion, if the amount of data needed is comparable to the size of the original dataset that would be a significant downside of the approach. Based on results presented we cannot say how much data is actually necessary to collect. In evaluation of some other online methods (like for example RaMP) we can clearly see how performance depends on the amount of interaction data collected.
> >
> > > (W4)
> >
> > I agree about the comparison being delicate. One issue is that the goal-conditioned method is given an arbitrarily more difficult task (removing goal labeling from half of the state space). No other method is given that variant, so it is unclear why comparing the two numbers would make any sense. One could choose any other percentage of goal labeling to be withheld to make the method arbitrarily worse.
> >
> > > (W5)
> >
> > I thank the authors for pointing out that the evaluation in Table 3 also requires trajectory stitching. I would suggest making this point in the main text of the paper. Section 6.5 being explicitly about trajectory stitching and not referring to those results, but to much simpler stitching evaluation, may cause the reader to miss the point.
> >
> > Additional evaluations on non-goal-reaching tasks are very much appreciated. I noticed that the performance of RaMP on "Stand" and "Jump" tasks seems lower than in the original paper (3767 ± 94 vs. ~5200 and 3098 ± 61 vs. ~4700), it is unclear why that is the case.
> >
> > > (Q1)
> >
> > Appreciate your response. I think that is a fair answer.
> >
> > > (Limitations)
> >
> > Once more, the additional evaluations are very much appreciated. Note/question about it in (W5) above.

---

> > > ### Author Response · Authors · 2024-08-13
> > >
> > > Thank you for responding to our rebuttal in detail. We address your questions below.
> > >
> > > (W1) There is indeed a balance between staying within the behavior distribution and seeking potentially more optimal behavior by going out of distribution. This holds true for both model-based and GOM-style methods. In fact, we found the guided diffusion planner generating out-of-distribution outcomes when the guidance coefficients is too large, hence the importance of tuning the guidance coefficient. Alternatively, we can train an ensemble of GOMs to quantify the epistemic uncertainty and add a penalty term to the planning cost to generate more in-distribution behavior. We do not claim GOMs to suffer from less limitations than model-based RL. Rather, they face the same tradeoff between optimality and accuracy, which can be addressed in a similar fashion.
> > >
> > > (W2) Thanks for clarifying your question. The overhead of random shooting @ 10 over guided diffusion comes from forwarding a 10 times larger batch through the diffusion sampling process and sorting the resulting values. The overhead is small per timestep, but accumulates over the number of diffusion and planning steps. We verify that planning with 1 random shooting sample (which is essentially sampling from the behavior policy) has roughly the same wall time as guided diffusion, at 42.5 seconds across 10 runs.
> > >
> > > We emphasize that the tradeoff between planners is not central to the contribution of GOMs. GOM is a versatile framework that is compatible with various planners, and the particular instantiation of GOM using diffusion models opens up the possibility of guided diffusion planner as a faster alternative to random shooting.
> > >
> > > We agree that the long-horizon planning ability is an important advantage of GOMs. We misunderstood your original question and conducted experiments with shorter planning horizons.
> > >
> > > (W3) In this paper, we focus on the offline setting, investigating the transfer behavior as a result of modeling capability. We evaluate our method and baseline in a controlled setting with the same offline dataset. The number of online samples required to infer the reward function is an intriguing research question, and one that we hope to explore in a future work. To speculate, given the same online interactions, our method would infer the same reward weights as RaMP, since both use linear regression for reward inference. The difference in reward inference, therefore, would lie in the exploration behavior of the learned policies.
> > >
> > > (W4) We realize the potential unfairness and confusion from introducing the misspecified goal-conditioned baseline. We are open to (1) presenting the full goal-conditioned baseline in Table 1 of the main paper, or (2) removing the goal-conditioned baseline from Table 1, keeping the comparison in Table 2. We hope to hear your suggestions on which would improve the paper more.
> > >
> > > (W5) We are glad to hear that you found the additional experiments valuable. In order to convert the Hopper task from RaMP to the GOM setting, we made two modifications: (1) the original version computes velocity by taking the finite difference of adjacent timesteps, which are not accessible from the state vector offline. We instead directly use the qvel from the state vector. (2) GOM require rewards to be a function of state, so we remove the action penalty from the reward. While this intuitively should increase the reward, the removal of action penalty can lead to behavior with higher variance. We will add these clarifications to the revision of the paper.
> > >
> > > We hope we have addressed your questions. Let us know if you have questions regarding our response.

---

### Author Rebuttal · Authors · 2024-08-07

We thank the reviewers for their careful reading and constructive feedback. We appreciate the reviewers for finding our approach “promising” and “novel, our paper “well-presented” and “easy to read,” and our experiments “encouraging.” We address some common questions here and defer more detailed responses to individual comments in the threads.
- We provide additional results on a new Hopper environment (Fig. 1) adapted from RaMP [1]. The environment features 4 tasks: forward, backward, stand, and jump, each with complicated reward function not directly specifiable by goals. We modify the original reward function by removing the action penalty, making the reward only dependent on the state. The dataset contains trajectories from replay buffers of expert policies trained to achieve each task. For GOM and USF, we use learned features pretrained with a dynamics prediction objective, as we found it to perform better than random features. As shown in Table 1 in the attached pdf, GOM compares favorable to representative baselines from each class of methods (USF, RaMP, COMBO), achieving the highest average return across 4 tasks.
- In Section 2, we perform additional ablation experiments.
    - For USF (Table 2), we ablate the choice of features and found that random Fourier features outperforms pretrained features with transition, Laplacian, or HILP [5] objectives.
    - For FB (Table 3), we ablate the feature dimensions and found decreasing the feature dimension to provide a regularization effect, although decreasing it too much restricts the representation capacity.
    - We ablate the effective planning horizon of GOM by reducing the discount factor $\gamma$ (Table 4). We found for shorter horizon tasks (antmaze-umaze), the optimality of planned trajectories improves as the planning horizon decreases. However, for long-horizon tasks (antmaze-medium), reducing the horizon too much hurts global optimality.
- Fig. 2 of the attached pdf visualizes rollouts of the GOM policy on the roboverse PickPlace and BlockedDrawer tasks. The dataset only contains separate trajectories completing the first phase (e.g. picking) or the second phase (e.g. placing). The GOM policy is able to perform trajectory stitching and generate a single trajectory to complete the task.
- Section 4 provides additional baseline comparisons on the preference antmaze task. USF and FB are able to distinguish different user preferences, although their performance is slightly worse than GOM. Another goal-conditioned learning baseline, GCSL [6], commits to one mode representing the shortest path, regardless of the user preference.
- Several reviewers raise questions about the notion of “policy-independence.” We clarify that GOM models the distributional successor feature of the behavior policy, which is not optimal for a specific reward, but covers the optimal policies for various rewards. For any reward function, GOM is able to find the best policy within the support of the dataset. This is contrasted with related works such as SF [2, 3] and gamma models [4], which model the occupancy of a single task-optimal policy. This means they cannot transfer to new rewards without redoing policy optimization.

References:

[1] Boyuan Chen, Chuning Zhu, Pulkit Agrawal, Kaiqing Zhang, Abhishek Gupta. RaMP: Self-Supervised Reinforcement Learning that Transfers using Random Features. NeurIPS 2023.

[2] André Barreto, Will Dabney, Rémi Munos, Jonathan J. Hunt, Tom Schaul, Hado van Hasselt, David Silver. Successor Features for Transfer in Reinforcement Learning. NeurIPS 2017.

[3] Harley Wiltzer, Jesse Farebrother, Arthur Gretton, Yunhao Tang, André Barreto, Will Dabney, Marc G. Bellemare, Mark Rowland. A Distributional Analogue to the Successor Representation. ArXiv 2024.

[4] Michael Janner, Igor Mordatch, Sergey Levine. Gamma-Models: Generative Temporal Difference Learning for Infinite-Horizon Prediction. NeurIPS 2020.

[5] Seohong Park, Tobias Kreiman, Sergey Levine. Foundation Policies with Hilbert Representations. ICML 2024.

[6] Dibya Ghosh, Abhishek Gupta, Ashwin Reddy, Justin Fu, Coline Manon Devin, Benjamin Eysenbach, Sergey Levine. Learning to Reach Goals via Iterated Supervised Learning. ICLR 2021.

---

### Decision · Program_Chairs · 2024-09-25

**Decision:**

Accept (poster)

**Comment:**

This paper proposes an approach to zero-shot reinforcement learning through learning the distribution of successor features in deterministic MDPs using offline data. It enables efficient computation of approximately optimal policies to achieve modeled outcomes. It can also generalize to new reward functions without additional environment interaction.
Empirical evaluation across numerous tasks shows strong performance and importantly  shows that the idea inspired by SFs can scale to more complex environments.
Strengths include potential reduction in error accumulation and flexibility with novel rewards.
Multiple reviewers agree to these evaluations and also applaud the clarity of writing.
However there were concerns involving, limitations in generating unseen outcomes, scalability concerns for complex environments, and assumptions about reward function accessibility. The evaluation methodology also raises some fairness and comprehensiveness questions.

There is a consensus inclination among all towards accept, with one reviewer being skeptical mainly due to limitations of baselines and experiments.  The rebuttal did a good job with clarifications and experiments.
The meta reviewer also read the paper in detail and finds that it is an interesting idea with sufficient technical  evidence to be of interest to a broader audience in policy learning.
The authors are advised to review the final comments, and update the manuscript accordingly.. In addition to including clarifications in the main paper, the authors are also advised to describe the possible limitations of this work in more detail.